# METAVLA: UNIFIED META CO-TRAINING FOR EFFICIENT EMBODIED ADAPTATION

**Chen Li**
Carnegie Mellon University
chenli4@andrew.cmu.edu

**Zhantao Yang**
Carnegie Mellon University
zhantaoy@andrew.cmu.edu

**Han Zhang**
Carnegie Mellon University
hanz3@andrew.cmu.edu

**Fangyi Chen**
Carnegie Mellon University
fangyic@andrew.cmu.edu

**Chenchen Zhu**
Meta Reality Labs, USA
chenchenz@meta.com

**Anudeepsekhar Bolimera**
Carnegie Mellon University
abolimer@andrew.cmu.edu

**Marios Saavides**
Carnegie Mellon University
marioss@andrew.cmu.edu

## ABSTRACT

Vision–Language–Action (VLA) models show promise in embodied reasoning, yet remain far from *true generalists*—they often require task-specific fine-tuning, incur high compute costs, and generalize poorly to unseen tasks. We propose **MetaVLA**, a unified, backbone-agnostic post-training framework for efficient and scalable alignment. MetaVLA introduces *Context-Aware Meta Co-Training*, which consolidates diverse target tasks into a single fine-tuning stage while leveraging structurally diverse auxiliary tasks to improve in-domain generalization. Unlike naive multi-task SFT, MetaVLA integrates a lightweight meta-learning mechanism—derived from Attentive Neural Processes—to enable rapid adaptation from diverse contexts with minimal architectural change or inference overhead. On the LIBERO benchmark, MetaVLA with six auxiliary tasks outperforms OpenVLA by up to 8.0% on long-horizon tasks, reduces training steps from 240K to 75K, and cuts GPU time by ∼76%. These results show that scalable, low-resource post-training is achievable—paving the way toward general-purpose embodied agents. Code will be available.

## 1 INTRODUCTION

Recent years have seen rapid progress in embodied Vision–Language–Action (VLA) models, which are typically pretrained from Vision–Language Models (VLMs) and adapted via supervised fine-tuning (SFT) (Kim et al., 2024; 2025; Hung et al., 2025) or reinforcement learning (RL) (Zhang et al., 2025; Li et al., 2025) to enable transfer to new embodiment tasks. In one line of work, a pretrained VLA backbone is adapted to autoregressively and discretely decode action tokens, trained on annotated demonstrations consisting of video or image observations paired with natural language instructions (Kim et al., 2024; Brohan et al., 2022; 2023; O'Neill et al., 2024). In contrast, another line of research represents output actions as continuous vectors, using techniques such as diffusion policies or flow matching (Black et al., 2024; Intelligence et al., 2025a; NVIDIA et al., 2025).

Despite advances in new task adaptation, current VLAs are not yet *true generalists*—still far from fully out-of-the-box usability and reliant on alignment through post-training (Zhou et al., 2025; Wang et al., 2025; Huang et al., 2025b; Din et al., 2025; Guruprasad et al., 2025; Ma et al., 2025). Compounding this, post-training remains practically constrained by benchmarks with low per-task data. Current practice (Kim et al., 2024) fine-tunes each downstream task independently, increasing overall training cost, hindering knowledge transfer across related tasks, and ultimately limiting success rate. These task-specific schedules are often brittle: many gradient steps are required before stably meaningful action sequences emerge, raising the risk of poor generalization and slowing adaptation to new task variants. For example, OpenVLA requires 240K training steps to fine-tune

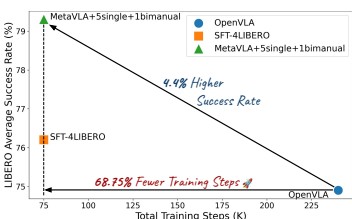 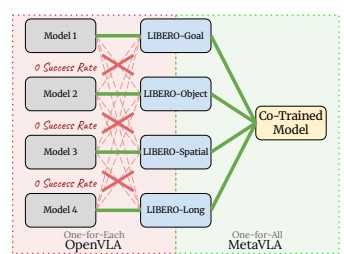 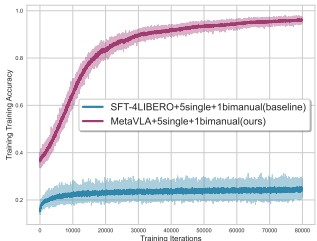

(a) **Higher success rate with fewer training steps.** MetaVLA achieves a 4.4% higher average success rate while requiring 68.75% fewer training steps compared to the OpenVLA baseline on LIBERO benchmarks.

(b) **Stronger cross-task generalization with one single model.** OpenVLA requires training separate models for each task suite, resulting in higher training costs and poor cross-task performance. In contrast, MetaVLA achieves strong generalization across all four suites with a single unified model.

(c) **Faster convergence to higher accuracy across all target tasks.** Comparison of training accuracy between MetaVLA and a baseline multi-task SFT when auxiliary tasks are added. MetaVLA consistently converges to higher accuracy across all LIBERO suites, while the baseline underperforms throughout training.

Figure 1: Three Key Merits of MetaVLA Compared to Baseline Approaches.

across all four LIBERO suites (OpenVLA Team, 2024), while OpenVLA-OFT (Kim et al., 2025) demands approximately 150K ~500K steps, including both diffusion and non-diffusion parts. Long-horizon tasks such as LIBERO-Long further dominate the training schedule and often become the system bottleneck.

While recent work (Black et al., 2024; Intelligence et al., 2025a; Qu et al., 2025) has focused on expanding datasets and exploring backbone architecture or training protocol innovations during pre-training, we instead tackle it from an orthogonal perspective at the post-training stage. Our experiments begin with a vanilla multi-task co-training setting: applying a standard SFT to a single model across related in-domain tasks (i.e., the four LIBERO suites). Indeed, we observe a reduction in total GPU training hours and improved success rates, which naturally motivates us to raise a question: can we introduce even more auxiliary tasks in the co-training to further boost VLA models? Sadly, we find that naively adding auxiliary tasks with greater domain diversity slows convergence and degrades performance. We attribute this surprise to the optimization instability arising from heterogeneous distributions, where misalignments in both the feature space (e.g., camera views) and action space (e.g., degrees of freedom) hinder the benefits of co-training.

Building on these ideas, we propose **MetaVLA**, a unified framework that fills a critical gap in VLA post-training by intelligently introducing auxiliary tasks without incurring the inefficiencies of per-task SFT or the performance drop of naive multi-task SFT. It introduces *Context-Aware Meta Co-Training*, which jointly trains all target tasks with a unified model, improving adaptation by leveraging cross-task data through a context bank. The context bank is a memory-augmented mechanism that contains auxiliary knowledge with domain diversities derived from Attentive Neural Processes (ANP) (Kim et al., 2019) based on *Meta-learning*. This lightweight module injects out-of-domain information gain without disrupting target optimization, enabling scalable and robust adaptation. MetaVLA is maintenance-friendly, backbone-agnostic, and easily extends beyond SFT to training paradigms like reinforcement learning. Figure 1 highlights three key advantages of MetaVLA over existing approaches.

Experiments show that MetaVLA with six auxiliary tasks outperforms the OpenVLA baseline by 4.4% and multi-task SFT by 3.1% on average, with gains up to 8.0% on LIBERO-Long. It unifies training into a single model, reducing steps from 240K to 75K and GPU time by 76%—from ∼100 to ∼24 hours. Despite its flexibility, the compact memory-augmented module adds only 0.3 ms/token in latency. The following sections present our framework, setup, and results, showing how MetaVLA boosts convergence, efficiency, and action reasoning. **Our main contributions are as follows**:

- We investigate an underexplored direction: improving post-training efficiency and generalization ability through incorporating diverse auxiliary tasks with negligible optimization overhead.

- We propose MetaVLA, a suite of plug-in module and training recipes that enables fast and scalable adaptation with strong generalization. MetaVLA is engineering-friendly and agnostic to backbone architectures and underlying training pipelines.

- We conduct comprehensive experiments to show that MetaVLA delivers superior performance with significant efficiency gains by reducing model count and GPU training hours, while preserving fast inference.

## 2 RELATED WORK

### 2.1 VISION-LANGUAGE-ACTION MODELS

Recent advances in Vision–Language–Action (VLA) models have been driven by supervised fine-tuning (SFT) of pretrained Vision–Language Models (VLMs) to map visual context and language instructions to action sequences—a stage we refer to as "pretraining" for VLA. These models are then adapted via SFT (Kim et al., 2024; 2025; Hung et al., 2025) or reinforcement learning (RL) (Zhang et al., 2025; Li et al., 2025) to unseen embodied tasks.

One line of work adapts pretrained VLA backbones to autoregressively decode discrete action tokens (Kim et al., 2024; Brohan et al., 2022; 2023; O'Neill et al., 2024), while another represents actions as continuous vectors using techniques like diffusion policies and flow matching (Black et al., 2024; Intelligence et al., 2025a; NVIDIA et al., 2025). For backbone design, recent studies explore alternatives such as Qwen2.5-VL (Bai et al., 2025; Qu et al., 2025; Hung et al., 2025). In parallel, efforts like EO-1 (Qu et al., 2025) introduce interleaved Vision-Text-Action training formats, while CoT-VLA (Zhao et al., 2025), OneTwoVLA (Lin et al., 2025) and ThinkAct (Huang et al., 2025a) incorporate reasoning data into training. Efficiency-focused works aim to improve VLA training through better tokenization or streamlined architectures (Pertsch et al., 2025; Reuss et al., 2025).

However, these approaches trade performance for costly pretraining interventions and meticulous data curation—an impractical strategy in resource-constrained or democratized settings. Moreover, achieving meaningful gains often requires careful design (Driess et al., 2025), incurring high human overhead. In contrast, our method operates entirely at the post-training stage, is orthogonal to existing techniques, and agnostic to both backbones and training pipelines—enabling seamless integration into various pretrained models and training recipes, including SFT and RL.

### 2.2 MULTI-TASK CO-TRAINING

Co-training across tasks has long been used to improve generalization (Doersch & Zisserman, 2017; Zhang & Yang, 2021), scalability (Devlin et al., 2019; Sun et al., 2020; McLean et al., 2025), and data efficiency (Aghajanyan et al., 2021; Crawshaw, 2020). More recently, it has shown strong success in LLMs and VLMs. GPT-2 (Radford et al., 2019), for example, leverages diverse pretraining sources (e.g., web pages, Wikipedia, news) for broad generalization. LLaVA (Liu et al., 2023b), a pioneering open-source VLM, uses multitask fine-tuning for multimodal alignment across conversation, captioning, and reasoning tasks. This trend continues in models like Qwen-3 (Yang et al., 2025), which expands co-training diversity by incorporating code, textbooks, and multilingual data across both pretraining and post-training. Similarly, Molmo and Pixmo (Deitke et al., 2024) provide detailed ablations on co-training with varied data sources, demonstrating the benefits of task and domain diversity. These advances highlight co-training as a key driver of performance in both pretraining and post-training stages.

Despite its effectiveness, co-training remains less explored in VLA, especially at post-training stage. While recent works (Kim et al., 2024; Team et al., 2024; Kim et al., 2025; Hung et al., 2025; Reuss et al., 2025) co-train during VLA pretraining, they afterwards still rely on task-specific fine-tuning for downstream adaption, missing the benefits of shared task structure for better generalization. This results in duplicated model checkpoints, costly maintenance, high total training steps and thus longer total GPU training hours. A few efforts have taken multi-task co-training for post adaption, but they

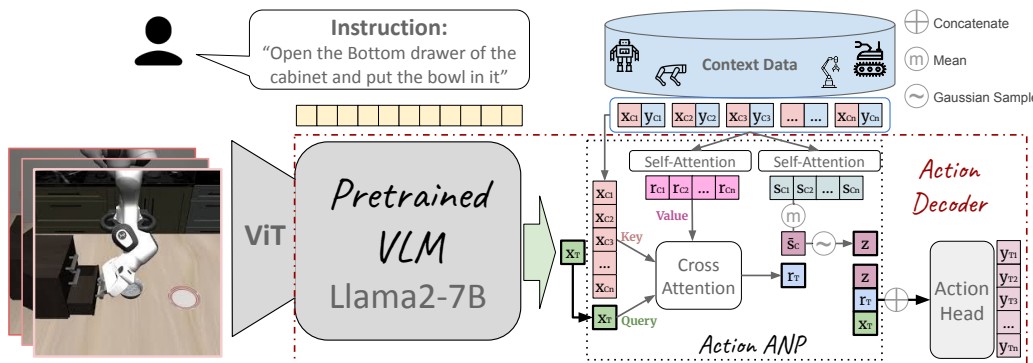

Figure 2: **MetaVLA Architecture**. VLA backbone married with *Context-Aware Meta Co-Training* Framework, where the context memory bank is composed of both in-domain target tasks and out-of-domain auxiliary tasks. We further detail the definitions of variables in Section 3.2.1.

are not free lunch. $\pi_0$ (Black et al., 2024) and $\pi_{0.5}$ (Intelligence et al., 2025a), require prohibitively expensive pretraining, while EO-1 (Qu et al., 2025) incurs high inference latency.

In contrast, our method systematically explores efficient task-shared adaptation in the post-training stage. It introduces a plug-and-play meta-learning module that enables scalable integration of unseen auxiliary tasks, enriching target learning with diverse signals. The approach is backbone-agnostic and streamlines efficient generalization across tasks, achieving strong performance gains via a lightweight, maintenance-friendly co-training paradigm.

## 2.3 META-LEARNING

Meta-learning enables models to quickly adapt to new tasks, often using diverse contextual data and through episodic training (Finn et al., 2017; Koch et al., 2015; Santoro et al., 2016; Ravi & Larochelle, 2016). Attentive Neural Processes (ANP) (Kim et al., 2019), an amortized meta-learners inspired by Gaussian Processes, learn a distribution over functions conditioned on both global prior and target-specific latent vectors via attention mechanism (Vaswani et al., 2023). ANP is well-suited for VLA due to its task-invariance, selective attention to relevant demonstrations, and avoidance of direct context optimization during adaptation. These properties simplify cross-domain training, enhance stability, and enable scalability—crucial for leveraging auxiliary data effectively, as shown in later results.

## 3 METHOD

### 3.1 TASK DEFINITION AND BACKBONE SELECTION

Our goal is to develop an efficient one-for-all VLA post-training paradigm capable of adapting to diverse novel tasks—unseen during pretraining.

Specifically, we adopt the LIBERO (Liu et al., 2023a) benchmark as our set of *target tasks* and use OpenVLA (Kim et al., 2024) as the backbone. Nevertheless, our method is backbone-agnostic and can be seamlessly integrated with other pretrained VLA models. See Section 4.1 for further details.

### 3.2 METAVLA

MetaVLA employs a *Context-Aware Meta Co-Training* approach that jointly trains on all in-domain suites with a single model, while leveraging contextual demonstrations through meta-learning. We formalize this mechanism by showing how *Action-ANP* offers a principled way to aggregate and condition on heterogeneous context data, using it to enhance stability during co-training across task suites.

### 3.2.1 ARCHITECTURE

To improve convergence and generalization in low-data task adaptation, we base our architecture on Attentive Neural Processes (ANP) (Kim et al., 2019)—a meta-learner inspired by Gaussian Processes that models a distribution over functions conditioned on both context and target representations. These latent codes capture global and task-specific semantics, aggregated via self-attention and cross-attention, respectively.

By attending to both current target data and related contextual data with ANP, the context data are aggregated and encoded into two separate branches: (1) a deterministic representation that captures target task dependent information by attending target tasks to context tasks; (2) a stochastic global representation that models information independent of target task based on context distribution. These two representations offer the model a referable demonstration for the current prediction.

We introduce a compact module, ***Action-ANP***, integrated into the Llama2 (Touvron et al., 2023) action decoder. Following the original ANP formulation, *Action-ANP* first applies self-attention (Vaswani et al., 2023) across context examples to extract a global prior, which is then fused with target queries through cross-attention (Vaswani et al., 2023) to form task-aware hybrid representations. Formally, given the target feature $x_T$, contextual feature-action pairs $(x_{Ci}, y_{Ci}) \in (x_C, y_C)$, *Action-ANP* models the conditional distribution of functions over target action $y_T$ given global and task-specific observations:

$$p(\mathbf{y}_T | \mathbf{x}_T, \mathbf{x}_C, \mathbf{y}_C) := \int p(\mathbf{y}_T | \mathbf{x}_T, \mathbf{r}_T, z)\, q(z | \bar{\mathbf{s}}_C)\, dz \tag{1}$$

Here, $\mathbf{r}_{Ci} \in \mathbf{r}_C$ and $\mathbf{s}_{Ci} \in \mathbf{s}_C$ are per-context representations aggregated from all contexts data pairs $(x_C, y_C)$ through self-attention. $r_T$ is the cross-attention output of query $x_T$ with context keys $x_{Ci}$ and values $\mathbf{r}_{Ci}$. $\bar{\mathbf{s}}_C$ is the mean of all $\mathbf{s}_{Ci}$, while $z$ is a stochastic latent drawn from the approximate posterior $q(z | \bar{\mathbf{s}}_C)$ computed over the context. During training, an additional condensed target representation $\bar{\mathbf{s}}_T$ is produced by the same self-attention and mean process as for $\bar{\mathbf{s}}_C$, with ground truth pair $(x_T, y_T)$. By reparameterizing the Gaussian latent $z$, the training objective maximizes a variational lower bound:

$$\log p(\mathbf{y}_T | \mathbf{x}_T, \mathbf{x}_C, \mathbf{y}_C) \geq \mathbb{E}_{q(z|\mathbf{s}_T)}[\log p(\mathbf{y}_T | \mathbf{x}_T, \mathbf{r}_T, z)] - D_{\mathrm{KL}}(q(z | \bar{\mathbf{s}}_T) \,\|\, q(z | \bar{\mathbf{s}}_C)) \tag{2}$$

This formulation enables MetaVLA to reconstruct target actions, regularized by a KL divergence that prevents the target distribution from drifting too far from the context distribution.

Unlike standard ANP, which uses smaller-scale neural networks, we integrate a pretrained Llama-2 (Touvron et al., 2023) backbone from OpenVLA. *Action-ANP* generates both stochastic and deterministic contextual latent vectors, which are concatenated with the Llama hidden states before the final output layer. The combined representations are then passed through the LM head to produce output logits, enabling end-to-end training via standard Llama decoding. See Figure 2 for an overview of the framework. We further summarize the symbols and definitions in Table 7.

### 3.2.2 DATA BANKS

In our setup, there are two data banks: context bank and target bank.

For context bank, which acts as an external memory, it's composed of both in-domain tasks, which are four LIBERO suites in our case, and auxiliary tasks. For in-domain tasks, the four LIBERO suites (Liu et al., 2023a) are split into non-overlapped context sets and target sets. For auxiliary tasks, we choose the large collection of partially open-sourced GR00T data (NVIDIA et al., 2025). A unified context bank then aggregates context sets from in-domain datasets and selected tasks from the auxiliary data. Details about auxiliary task selection will be discussed in Section 3.3.

The target data bank contains only the target sets of in-domain tasks—in our case, the task sets across all four LIBERO suites. Unlike standard VLA SFT, which trains a separate model for each suite, our meta co-training strategy trains a single model across all target suites, improving scalability, generalization, and efficiency.

### 3.2.3 TRAINING PROTOCOLS

To ensure broad contextual coverage, we refresh the context set every $\mathbf{K}$ training steps. Specifically, at each multiple of $\mathbf{K}$, we randomly sample $\mathbf{b}_C$ examples from each context task's dataset, keeping $\mathbf{b}_C$ consistent across tasks for simplicity. We set $\mathbf{K} = 200$ to balance training speed and decoding quality, and choose $\mathbf{b}_C = 32$ to optimize memory usage and performance. An ablation study on $\mathbf{b}_C$ is provided in Section 4.4.2.

## 3.3 AUXILIARY TASKS SELECTION

To enhance context diversity and strengthen meta-learning, we introduce an auxiliary task selection mechanism. Specifically, we incorporate the GR00T dataset (NVIDIA et al., 2025; NVIDIA, 2025) into the context bank for two key reasons. First, GR00T is entirely unseen during OpenVLA pretraining, making it a valuable source of additional information gain. Second, it offers partial domain relevance to LIBERO while differing structurally—striking a balance between familiarity and diversity.

LIBERO tasks feature a Franka Emika Panda arm with a gripper and primarily use front-facing camera views. In contrast, selected GR00T tasks include bimanual manipulation using front views and single-arm manipulation with side views only. These variations are intentionally chosen to test the robustness and generalization ability of MetaVLA. An example of task difference among these tree types is showing in Figure 3, and more examples are in Section A.2.

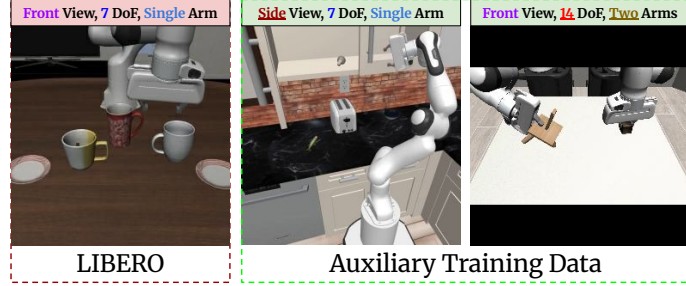

Figure 3: **Comparison between auxiliary tasks and LIBERO evaluation benchmark.** LIBERO tasks use third-person front-view images and 7-DoF actions for a single-arm robot. In contrast, our auxiliary data from GR00T introduces variation through side-view observations and a two-arm robot with 14-DoF actions. MetaVLA benefits from this data diversity, while Open-VLA struggles with the domain mismatch.

Unlike Zhao et al. (2025), which carefully select tasks highly similar to LIBERO, our method is less strict in data varieties in the context bank, and more robust to the diversity of auxiliary tasks which we believe would introduce higher freedom for a more scalable adaption training framework. Experimental results show that MetaVLA, equipped with this multi-task co-training setup, achieves higher success rates and faster convergence across all LIBERO suites compared to vanilla SFT-based co-training. Ablation study on the effect of auxiliary task selection is presented in Section 4.4.3.

## 4 EXPERIMENTS

### 4.1 EXPERIMENT SETTING

We evaluate our method against previous works on the LIBERO benchmark (Liu et al., 2023a), a Franka Emika Panda single arm simulation-based benchmark with four different task suites. The benchmark aims to evaluate the model's capability of generalizing to variations of the 500 expert demonstrations across 10 tasks provided for each task suite. **LIBERO-Goal** leaves objects and layouts unchanged, and varies by final task goals; **LIBERO-Spatial** keeps the objects and tasks unchanged, while re-arranging the layout; **LIBERO-Object** uses the same layout environment, while changing the object types; **LIBERO-Long** (also known as LIBERO-10) consists of long horizon tasks with a mixture of different distribution shifts above. Our method co-trains a single model for all four suites with up to 6 heterogeneous auxiliary tasks with panda gripper robots from GR00T dataset (NVIDIA, 2025), a simulation dataset consisting of different robots and task types, More details are discussed in Section 3.2.2, 3.3, and A.2. We follow prior work (Kim et al., 2024; In-

| Model | Steps | Goal | Spatial | Object | Long | Average |
|---|---|---|---|---|---|---|
| $\pi_{0.5}$ (Intelligence et al., 2025b) | 30K | 98.0 | 98.8 | 98.2 | 92.4 | 96.9 |
| Diffusion Policy (Chi et al., 2023) | - | 68.3 | 78.3 | **92.5** | 50.5 | 72.4 |
| ATM (Wen et al., 2023) | - | 77.8 | 68.5 | 68.0 | 39.3 | 63.4 |
| TraceVLA (Zheng et al., 2025) | - | 75.1 | 84.6 | 85.2 | 54.1 | 74.8 |
| OpenVLA (Kim et al., 2024) | 240K | 76.2 | 84.7 | 87.0 | 51.8 | 74.9 |
| SFT-4LIBERO | | 77.8 | 84.8 | 87.4 | 54.7 | 76.2 |
| SFT-4LIBERO+1single+1bimanual | | 59.7 | 68.0 | 65.2 | 30.0 | 55.7 |
| SFT-4LIBERO+3single | 75K | 24.6 | 16.8 | 9.7 | 1.5 | 13.2 |
| SFT-4LIBERO+5single+1bimanual | | 15.2 | 5.6 | 12.0 | 1.6 | 8.6 |
| SFT-4LIBERO+5single+1bimanual | 187.5K | 23.4 | 16.7 | 13.6 | 4.4 | 14.5 |
| MetaVLA-Pretrained-Context-ONLY | | 74.4 | 85.4 | 85.4 | 52.3 | 74.4 |
| MetaVLA (ours) | | **78.9** | 88.5 | 88.5 | 55.3 | 77.8 |
| MetaVLA+Stochastic (ours) | 75K | **78.9** | 88.9 | 88.5 | 53.0 | 77.3 |
| MetaVLA+1single+1bimanual (ours) | | 78.5 | 89.0 | 87.4 | 59.0 | 78.5 |
| MetaVLA+3single (ours) | | 78.0 | 88.0 | 87.2 | 59.7 | 78.2 |
| MetaVLA+5single+1bimanual (ours) | | 78.7 | **89.9** | 88.9 | **59.8** | **79.3** |

Table 1: **Success rate (%) and training step comparison with prior methods.** All MetaVLA variants are single models trained for 75K steps. *MetaVLA (ours)* uses only LIBERO suites in the context bank without the stochastic module, while *MetaVLA+Stochastic (ours)* includes it. *Method+NSingle+Mbimanual* includes $N$ single arm and $M$ bimanual (two arms) auxiliary tasks described in Section 3.3. *SFT-4LIBERO* is a single-model baseline trained with vanilla multi-task SFT across all suites. *OpenVLA (top)* comprises four Hugging Face models fine-tuned separately on LIBERO using the OpenVLA-7B backbone, totaling roughly 240K steps. MetaVLA with six auxiliary tasks surpasses OpenVLA by 4.4% and SFT-4LIBERO by 3.1% on average, with even larger gains on LIBERO-Long (8.0% and 5.1%, respectively).

telligence et al., 2025b) and adopt *Success Rate* (SR) as our evaluation metric. Thanks to efficient co-training, our method requires only ∼24 hours to fine-tune across all four LIBERO suites using 8 A100 80GB GPUs. We use OpenVLA (Kim et al., 2024) as our backbone due to its completeness, maturity, and robust open-source codebase and evaluation pipeline, which has been widely adopted in the academic community.

To ensure fair comparison, we re-evaluate the OpenVLA baselines in the LIBERO simulation environment using the four single-task fine-tuned models from Hugging Face (OpenVLA Team, 2024), and adopt these as our baselines. Due to hardware variance and stochasticity, our results may slightly differ from the originally reported values (OpenVLA Contributors, 2024). All the reported results on LIBERO are evaluated on one 24GB RTX-4090 GPU.

## 4.2 EFFECT OF VANILLA MULTI-TASK SFT

As shown in Table 1, adding auxiliary tasks to vanilla multi-task SFT (**SFT-4LIBERO+*auxiliary tasks***) consistently degrades performance. The degradation worsens as more tasks are added, suggesting the model struggles with domain shifts and fails to converge. Training convergence curves in Figure 5 further corroborate this finding. One possible factor is reduced training steps per task. For instance, in **SFT-4LIBERO+5single+1bimanual** trained for 75K steps, per-task steps drop from 18.75K (in SFT-4LIBERO) to 7.5K. To test this, we increase training to 187.5K steps. While performance improves slightly, it remains well below MetaVLA—with or without auxiliary tasks. Furthermore, as shown in Figure 6, training curves at 187.5K steps across all three metrics—Accuracy, Imitation Loss, and L1 Loss—signal its suboptimal adaptation. This supports our view that MetaVLA scales more robustly, leveraging auxiliary data without encountering optimization instability. A more rigorous proof of this view is left to future work due to computational constraints.

### 4.3 Effect of Context-Aware Meta Co-Training

As shown in Table 1, MetaVLA—with or without auxiliary tasks—outperforms all baselines, including OpenVLA baseline and SFT-4LIBERO, across all LIBERO tasks and on average. With six auxiliary tasks, it improves over OpenVLA by 4.4% and SFT-4LIBERO by 3.1%, **notably on LIBERO-Long**, with gains of 8.0% and 5.1%, respectively. Moreover, MetaVLA reduces model count to one and cuts training steps from 240K to 75K. Examples of success cases are demonstrated in Section A.6.

### 4.4 Ablation Study

#### 4.4.1 Effect of Different Backbone

| Model | Goal | Spatial | Object | Long | Average |
|---|---|---|---|---|---|
| NORA-Long (Hung et al., 2025) | 85.4 | 90.5 | 95.0 | 70.6 | 85.4 |
| NORA-Long-SFT-4LIBERO | 87.0 | 92.5 | 94.0 | 75.5 | 87.3 |
| NORA-Long-SFT-4LIBERO+5single+1bimanual | 73.6 | 79.5 | 75.2 | 37.2 | 66.4 |
| MetaVLA-NORA-Long (ours) | 90.8 | **96.2** | 96.5 | 77.8 | 90.3 |
| MetaVLA-NORA-Long+5single+1bimanual (ours) | **93.8** | 95.8 | **97.2** | **80.2** | **91.8** |

Table 2: **Success rate comparison of applying MetaVLA variants to NORA-Long.** *NORA-Long-SFT-4LIBERO* is a single-model baseline trained with vanilla multi-task SFT across all suites using NORA-Long. *NORA-Long* comprises four Hugging Face models fine-tuned separately on LIBERO using the NORA-Long backbone. Without auxiliary tasks, MetaVLA-NORA-Long surpasses NORA-Long by 4.9%, and exceeds NORA-Long-SFT-4LIBERO by 3.0%. Adding auxiliary tasks further pushes the average success rate to 91.8%, achieving the highest performance in LIBERO-Goal, Object, and Long.

To validate MetaVLA's effectiveness in various backbones, we assessed MetaVLA variants on NORA (Hung et al., 2025), a 3B Qwen2.5-VL-based (Bai et al., 2025) VLA model. We selected NORA-Long (Deep Cognition and Language Research (DeCLaRe) Lab, 2025) variant as the base model because it provides a stronger LIBERO performance baseline than NORA. To ensure fair comparison, we re-evaluate the NORA-Long baselines in the LIBERO simulation environment using the four single-task fine-tuned models from Hugging Face (Deep Cognition and Language Research (DeCLaRe) Lab, 2025), and adopt these as our baselines.

As shown in Table 2, without auxiliary data, MetaVLA outperforms NORA-Long by 4.9% on average and NORA-Long-SFT-4LIBERO by 3.0%. When auxiliary tasks are incorporated, the average success rate further improves by 6.4% compared to NORA-Long. For more challenging suites—**Goal** and **Long**—the improvements are 8.4% and 9.6%, respectively. Moreover, consistent with results using the OpenVLA backbone, **MetaVLA-NORA-Long+5single+1bimanual** significantly outperforms its native SFT counterpart, **NORA-Long-SFT-4LIBERO+5single+1bimanual**, by **25.4%**. The training convergence curves on accuracy and loss in Figure 10 further bolster the stronger stability of MetaVLA during training as more diverse tasks are added. Together, these results demonstrate MetaVLA's backbone-agnostic capability.

#### 4.4.2 Effect of Context Batch Size

As shown in Figure 4, success rate increases monotonically with batch size under our setting. A relatively small context batch size of 32 yields the best performance, which doesn't introduce extra overhead to memory footprint. A detailed table is shown in Table 6 in Appendix.

#### 4.4.3 Effect of Auxiliary Task Selection

As shown in Table 1, MetaVLA outperforms its SFT-4-LIBERO counterparts across all three auxiliary task settings, demonstrating robust generalization to variations in camera views, action spaces, and the number of context tasks. These results highlight a promising opportunity to scale up the context bank.

Figure 4: **Left: Per-suite LIBERO success rate across varying context batch sizes.** *OpenVLA* refers to the four Hugging Face baseline models, each fine-tuned individually on LIBERO using the OpenVLA-7B backbone, while *SFT-4LIBERO* is a single-model baseline trained with vanilla multi-task SFT across all suites. For each suite, success rate increases monotonically with context batch size. **Right: Average success rate across LIBERO suites with varying context batch sizes.** *OpenVLA* denotes the four Hugging Face models baselines fine-tuned individually on LIBERO with the OpenVLA-7B backbone, while *SFT-4LIBERO* is a single-model baseline trained with vanilla multi-task SFT across all suites. $b_c$ indicates the context batch size. Larger context batches consistently yield higher average success rates.

### 4.4.4 EFFECT OF PARAMETER SIZE CHANGE

To rule out the possibility that performance gains stem solely from increased parameter size, we conduct an ablation in which the architecture remains unchanged, but the context bank is replaced to include only tasks—*bridge_orig* and *fractal20220817_data*—both part already included in the OpenVLA pretraining dataset (OpenVLA Contributors, 2024). The result, denoted as **MetaVLA-Pretrained-Context-ONLY** in Table 1, shows a significant drop across all LIBERO suites compared to MetaVLA. This suggests that the performance boost is not simply due to increased parameter size, but rather stems from the full design portfolio along with the integration of ***exotic auxiliary tasks*** that enrich the context with diverse and informative signals.

### 4.4.5 EFFECT OF MULTI-TASK CO-TRAINING MECHANISM

To assess the impact of task-shared co-training, we replace MetaVLA's full target set (all four LIBERO suites) with a single suite at a time. For simplicity, we adopt a frugal context bank containing only the four LIBERO suites without auxiliary tasks—matching the setup in Table 1 for **MetaVLA**. Under this setting, we train four models independently via SFT, one per suite, using the same total training steps (240K) as OpenVLA (OpenVLA Team, 2024). We refer to this configuration as **MetaVLA-EACH**. For evaluation, we report results for both OpenVLA baselines and MetaVLA-EACH at 240K (final step) and 120K (mid-training) to highlight the earlier convergence benefits of MetaVLA.

Results in Table 3 reveal three key findings: (1) **MetaVLA-EACH** outperforms the Hugging Face OpenVLA baselines (OpenVLA Team, 2024) at final steps; (2) it achieves higher success rates earlier in training across all suites; and (3) on complex suites (Goal, Long), performance continues to improve, while simpler ones (Spatial, Object) converge earlier—suggesting that task diversity benefits more challenging tasks.

These findings highlight the effectiveness of *Action-ANP* within a scalable, memory-based meta-learning framework. However, compared to full **MetaVLA** (Table 1), MetaVLA-EACH sacrifices unified generalization and training efficiency, requiring four models and more compute (120K vs. 75K steps).

### 4.4.6 EFFECT OF STOCHASTIC LEARNING

As shown in the ELBO bound equation 2, *Action-ANP* jointly optimizes a reconstruction loss and a KL divergence term. In Table 1, **MetaVLA+Stochastic** includes this stochastic regularization, while **MetaVLA** does not. The stochastic variant improves performance on the Spatial suite, performs comparably on Goal and Object, but underperforms on Long. Since the KL term encourages proximity between context and target distributions—an assumption that may not hold in more complex settings—we hypothesize that the greater domain shift in Long tasks leads to this performance

| Method | Total Steps | Steps | Goal | Steps | Spatial | Steps | Object | Steps | Long |
|---|---|---|---|---|---|---|---|---|---|
| OpenVLA-120K (Kim et al., 2024) | 120K | 30K | 71.4 | 10K | 81.2 | 30K | 85.8 | 50K | 44.4 |
| MetaVLA-EACH-120K | 120K | 30K | 76.4 | 10K | **86.1** | 30K | **89.0** | 50K | 55.4 |
| OpenVLA-240K (Kim et al., 2024) | 240K | 60K | 76.2 | 50K | 84.7 | 50K | 87.0 | 80K | 51.8 |
| MetaVLA-EACH-240K | 240K | 60K | **77.4** | 50K | 85.8 | 50K | 88.5 | 80K | **55.8** |

Table 3: **MetaVLA-EACH: Per-suite success rates across LIBERO.** *OpenVLA* denotes the four baseline models fine-tuned separately for each LIBERO suite, released on Hugging Face and trained for 240K total steps. *OpenVLA-120K* follows the same setup but with 120K steps. *MetaVLA-EACH-120K* and *MetaVLA-EACH-240K* are our models trained separately per suite for 120K and 240K steps, respectively, without co-training. Thanks to the *Action-ANP* design, all MetaVLA-EACH variants outperform their OpenVLA counterparts with fewer steps. For Goal and Long, performance continues to improve at 240K steps, indicating stronger learning potential.

drop. In contrast, the deterministic variant, which relies solely on reconstruction loss, provides more precise modeling, making it more effective for challenging tasks. For this reason, the stochastic module is disabled in all other MetaVLA experiments for practicality.

## 4.5 EFFICIENCY DISCUSSION

We evaluate all tasks using one RTX-4090 GPU with batch inference. Our method added slightly more trainable parameters to the original architecture due to its lightweight property, which only increases inference latency by 0.3 ms/token, shown in Figure 9 in Appendix. Moreover, it reduces total GPU training time by 76%—from ∼100 to ∼24 hours—by cutting training steps from 240K to 75K. It also consolidates four task-specific models into a single one, streamlining deployment and improving maintenance efficiency.

## 4.6 WHY DOES OUR METHOD WORK?

Multi-task co-training promotes knowledge sharing across related in-domain tasks, while *Action-ANP* leverages diverse auxiliary data to boost target performance and mitigate optimization instability from domain shifts. As shown in Figure 5, MetaVLA consistently outperforms naive multi-task SFT across all three convergence metrics—Accuracy, Imitation Loss, and L1 Loss. The first two assess the quality of generated discrete tokens, while L1 Loss measures the resulting continuous actions for robot execution. These results demonstrate both the effectiveness and stability of our approach.

In Section 4.4.2, we observe a monotonic performance gain with larger context batch sizes, and in Section 4.4.3, a steady improvement with more diverse auxiliary tasks. While we do not exhaust all combinations due to memory and compute constraints, these trends suggest the potential of ***Context Scaling***—increasing batch size and task diversity in the context bank may further enhance target-task performance. Moreover, given MetaVLA's robustness to context diversity, augmenting the context bank with web-scale data—previously explored only at the pretraining stage (Black et al., 2024; Intelligence et al., 2025a; Qu et al., 2025)—may offer additional benefits. We leave this to future work.

## 5 CONCLUSION

We presented **MetaVLA**, a framework that addresses the inefficiencies and brittleness of current VLA post-training pipelines. By introducing *Context-Aware Meta Co-Training*, MetaVLA integrates auxiliary tasks without destabilizing optimization, achieving superior convergence speed, efficiency, and generalization. MetaVLA is lightweight, plug-and-play, and backbone-agnostic, making it easy to extend beyond supervised fine-tuning to reinforcement learning or hybrid pipelines. Empirical results on LIBERO show consistent gains over both per-task fine-tuning and naive multi-task SFT, while significantly reducing training cost and model count. Looking forward, we envision extending MetaVLA to broader backbones and wider benchmarks, incorporating web-scale multimodal data, and deploying on real robots. We hope this work inspires future research toward efficient, scalable, and truly generalist embodied VLA systems.

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

## A APPENDIX

### A.1 TRAINING CONVERGENCE

Figure 5 presents Training Accuracy, Imitation Loss (cross-entropy over generated discrete action tokens), and L1 Loss (on the transformed continuous actions) for three auxiliary task settings: 1single+1bimanual, 5single+1bimanual, and 3single. In all cases, MetaVLA consistently converges to higher performance across all three metrics.

### A.2 CONTEXT TASK DETAILS

We use the LIBERO dataset (Liu et al., 2023a) as both target and context tasks, and GR00T (NVIDIA, 2025) as auxiliary context tasks only. A detailed breakdown of the datasets is provided in Table 4. Example tasks from LIBERO and GR00T are visualized in Figures 7 and 8, respectively.

| Dataset | Tasks |
|---|---|
| LIBERO (Liu et al., 2023a) | LIBERO-Goal
LIBERO-Spatial
LIBERO-Object
LIBERO-Long |
| GR00T (NVIDIA, 2025) | bimanual_panda_gripper.Threading
single_panda_gripper.CoffeeServeMug
single_panda_gripper.OpenDrawer
single_panda_gripper.PnPCabToCounter
single_panda_gripper.PnPCounterToMicrowave
single_panda_gripper.TurnSinkSpout |

Table 4: **Summary of datasets and tasks used in the experiments.**

### A.3 MODEL ARCHITECTURE AND TRAINING DETAILS

**Model Architecture** We build on OpenVLA-7B (Kim et al., 2024) as the base model, integrating *Action-ANP*, a lightweight, memory-based meta-learning module. In *Action-ANP*, global prior representations are encoded via self-attention, while cross-attention (Vaswani et al., 2023) fuses target and context to produce a final hybrid latent representation. Each attention block is followed by Layer Normalization and a final MLP projection.

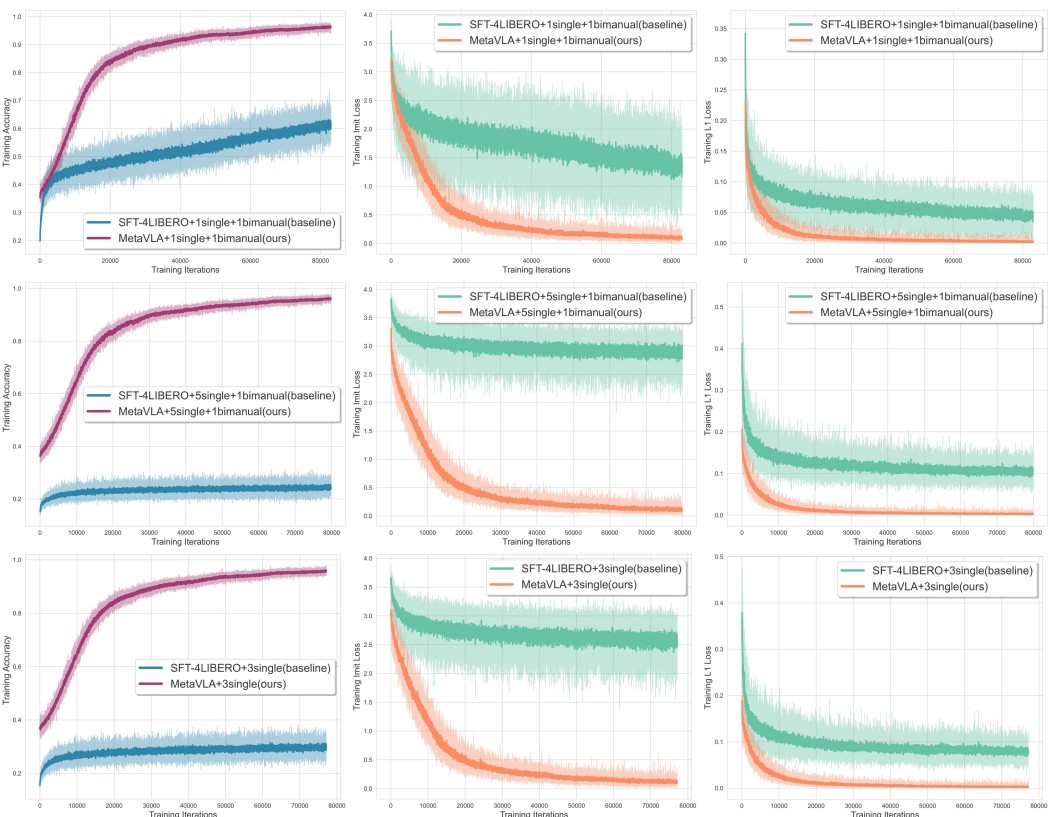

Figure 5: **Training convergence comparison for models trained with 75K steps.** Training Accuracy, Imitation Loss, and L1 Loss are compared between *MetaVLA* variants and *SFT-4LIBERO* under different auxiliary-task settings. All *MetaVLA* variants consistently converges to superior performance across all three metrics, while *SFT-4LIBERO* fails to adapt effectively—highlighting the robustness and scalability of our approach.

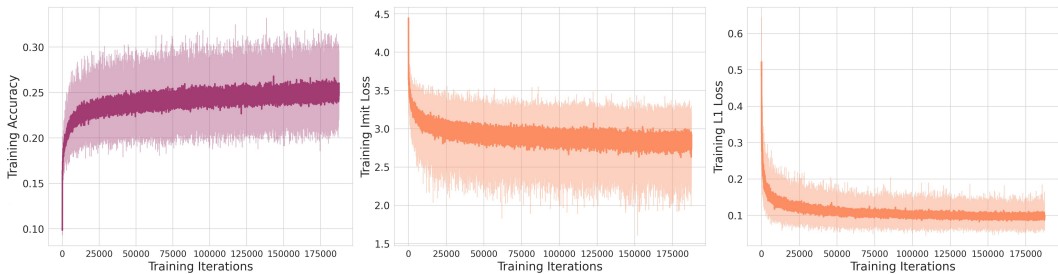

Figure 6: Training convergence of MetaVLA with six auxiliary tasks (one bimanual and five single-arm) trained with 187.5K steps. All three metrics—Accuracy, Imitation Loss, and L1 Loss—converge to suboptimal levels.

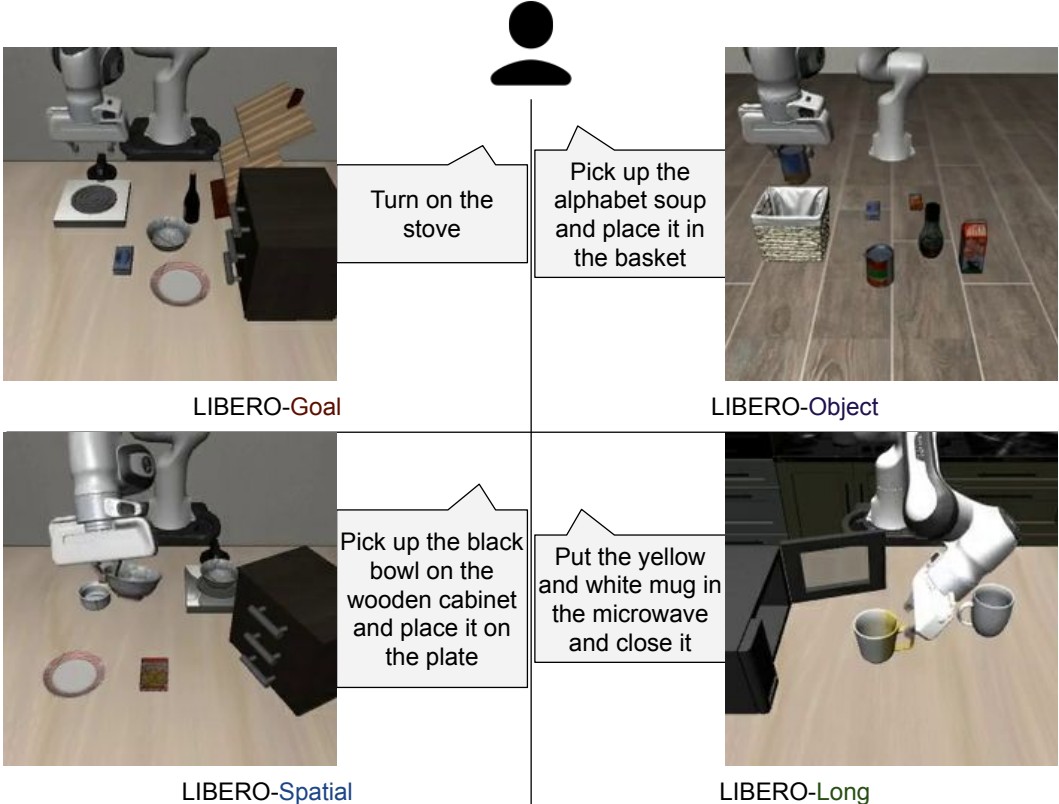

Figure 7: **LIBERO examples.** Each suite example includes a frame from the primary camera view together with its task instruction.

**Training Settings** We trained all MetaVLA variants with LoRA (Hu et al., 2021) on 8 A100-80 GB GPUs with 75K training steps, taking approximately 24 GPU hours, using 8 x 80GB VRAM. Training hyperparameters are in Table 5.

## A.4 EXPERIMENT DETAILS

### A.4.1 INFERENCE EFFICIENCY

Our method is engineering-friendly and computationally lightweight. We measure both token throughput and latency of the model end-to-end, on one 24GB RTX-4090 GPU against Open-VLA (OpenVLA Contributors, 2024). All environments and packages are kept the same throughout the experiment to ensure fair comparison. Our efficiency results are shown in Figure 9. Our

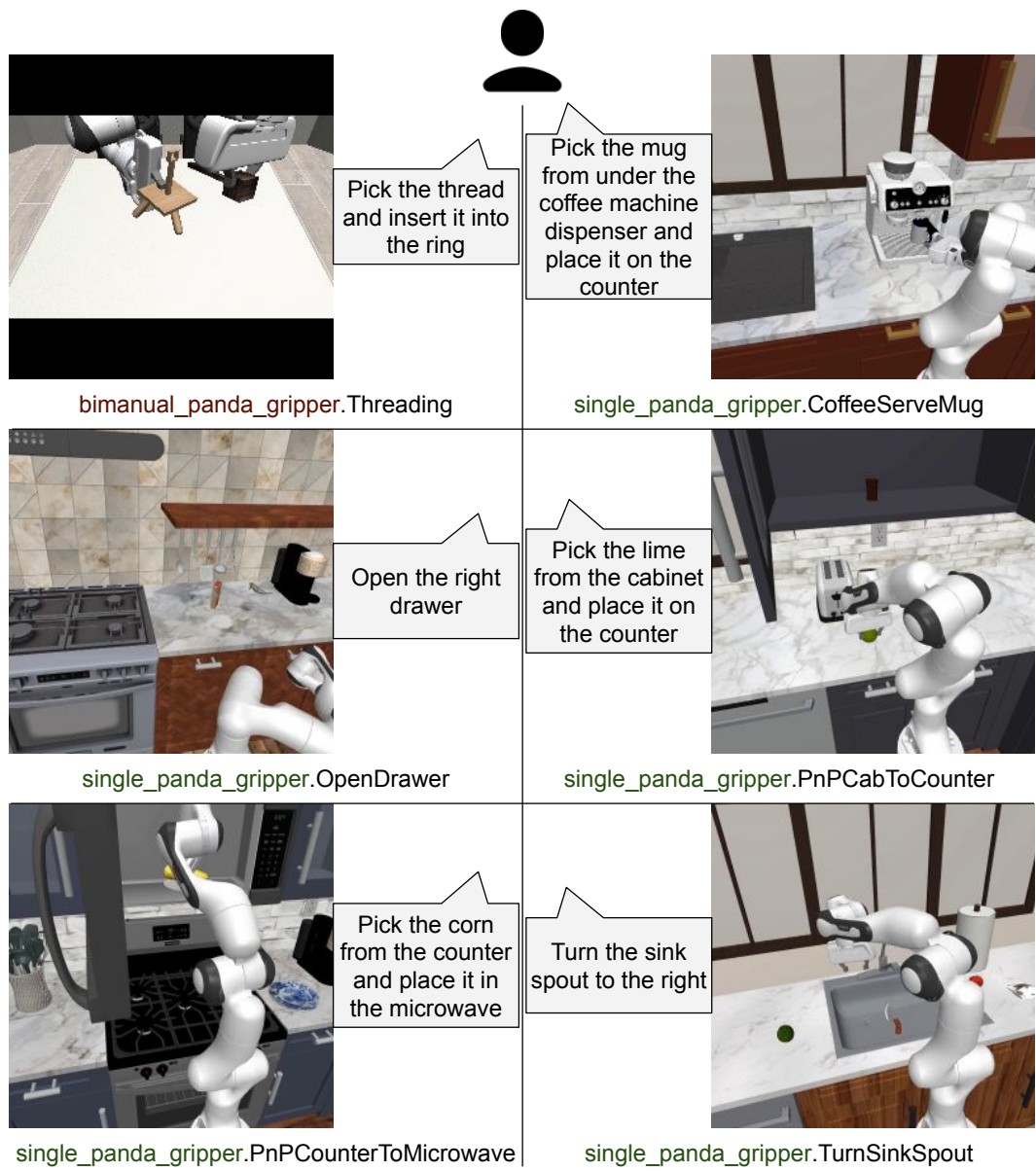

Figure 8: **GR00T examples.** Each task example includes a frame from the primary camera view paired with its task instruction.

| Hyperparameter | Value |
|---|---|
| Shuffle Buffer Size | 100000 |
| FlashAttention-2 | Enabled |
| LoRA Rank | 32 |
| LoRA Dropout | 0.0 |
| Total Batch Size | 128 |
| Gradient Accumulation Steps | 1 |
| Learning Rate | 5e-4 |
| Context Batch Size | 32 |
| *Action-ANP* Latent Dimension | 2048 |

Table 5: **Training Hyperparameters.** Total batch size is computed as 16 samples per GPU across 8 GPUs. Context batch size refers to the batch size used for each individual context task.

*Action-ANP* module introduces approximately 5.5% more latency compared to OpenVLA, making MetaVLA an ideal practical choice for achieving a higher success rate.

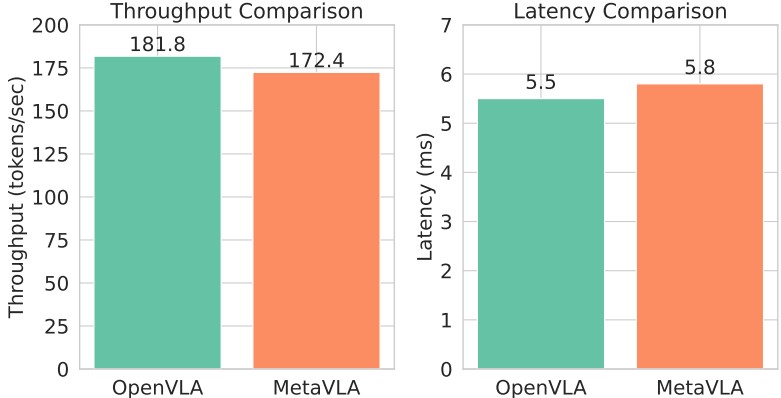

Figure 9: **Efficiency Metrics.** Our lightweight module only adds negligible overhead to inference cost, making MetaVLA practical for deployment and usage.

### A.4.2 EFFECT OF CONTEXT BATCH SIZE

Table 6 shows the success rates of MetaVLA across different LIBERO tasks using different context batch sizes. The performance scales up as we introduce more contextual data.

| Method | Goal | Spatial | Object | Long | Average |
|---|---|---|---|---|---|
| OpenVLA (OpenVLA Contributors, 2024) | 76.2 | 84.7 | 87.0 | 51.8 | 74.9 |
| SFT-4LIBERO | 77.8 | 84.8 | 87.4 | 54.7 | 76.2 |
| MetaVLA($\mathbf{b}_C = 4$) | 75.0 | 82.2 | 85.0 | 50.4 | 73.2 |
| MetaVLA($\mathbf{b}_C = 8$) | 75.4 | 85.5 | 86.8 | 51.4 | 74.8 |
| MetaVLA($\mathbf{b}_C = 16$) | 76.8 | 87.8 | 88.0 | 54.3 | 76.7 |
| MetaVLA($\mathbf{b}_C = 32$) | **78.9** | **88.5** | **88.5** | **55.3** | **77.8** |

Table 6: Effect of different context batch sizes across different LIBERO task suites.

### A.4.3 EFFECT OF DIFFERENT BACKBONE

Figure 10 shows Training Accuracy and Imitation Loss for 5 single and 1 bimanual auxiliary tasks when using NORA-Long (Hung et al., 2025) backbone. MetaVLA consistently converges to higher performance across both metrics compared to vanilla SFT, proving that our observation on Open-VLA (OpenVLA Contributors, 2024) is also true under another backbone.

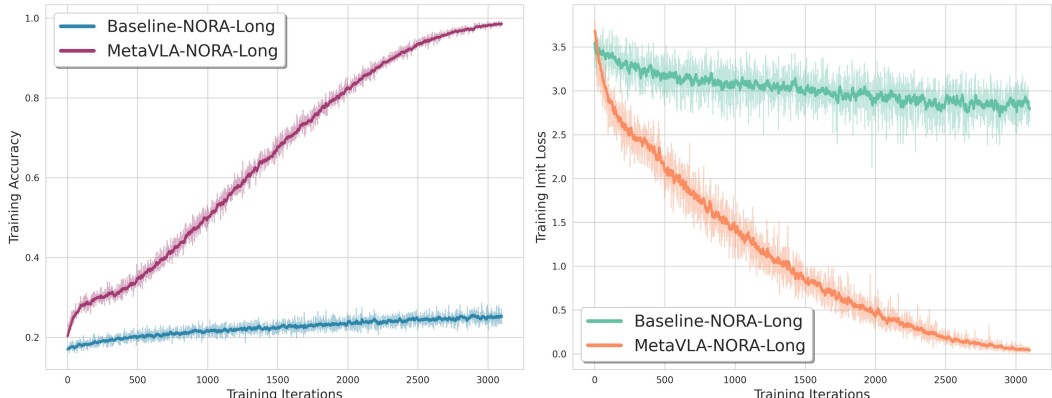

Figure 10: **Training convergence comparison for NORA-Long backbone with and without MetaVLA after adding 5 single and 1 bimanual auxiliary tasks.** Training Accuracy and Imitation Loss are compared between MetaVLA variants and baseline *SFT-4LIBERO* under *5single+1bimanual* co-training settings. Without *MetaVLA*, vanilla SFT with auxiliary tasks fails to adapt effectively, while proposed *MetaVLA* consistently achieves better accuracy and lower loss throughout the training.

### A.5 SYMBOLS AND DEFINITIONS

We summarize all the symbols used in our MetaVLA architecture in Table 7

### A.6 SUCCESS CASES IN LIBERO SIMULATION

Figures 11, 12, 13, and 14 demonstrate example execution sequences of MetaVLA successfully completing one task from each LIBERO suite in its simulation: Goal, Spatial, Object, and Long.

### A.7 LLM USAGE

We used LLM to aid and polish writing.

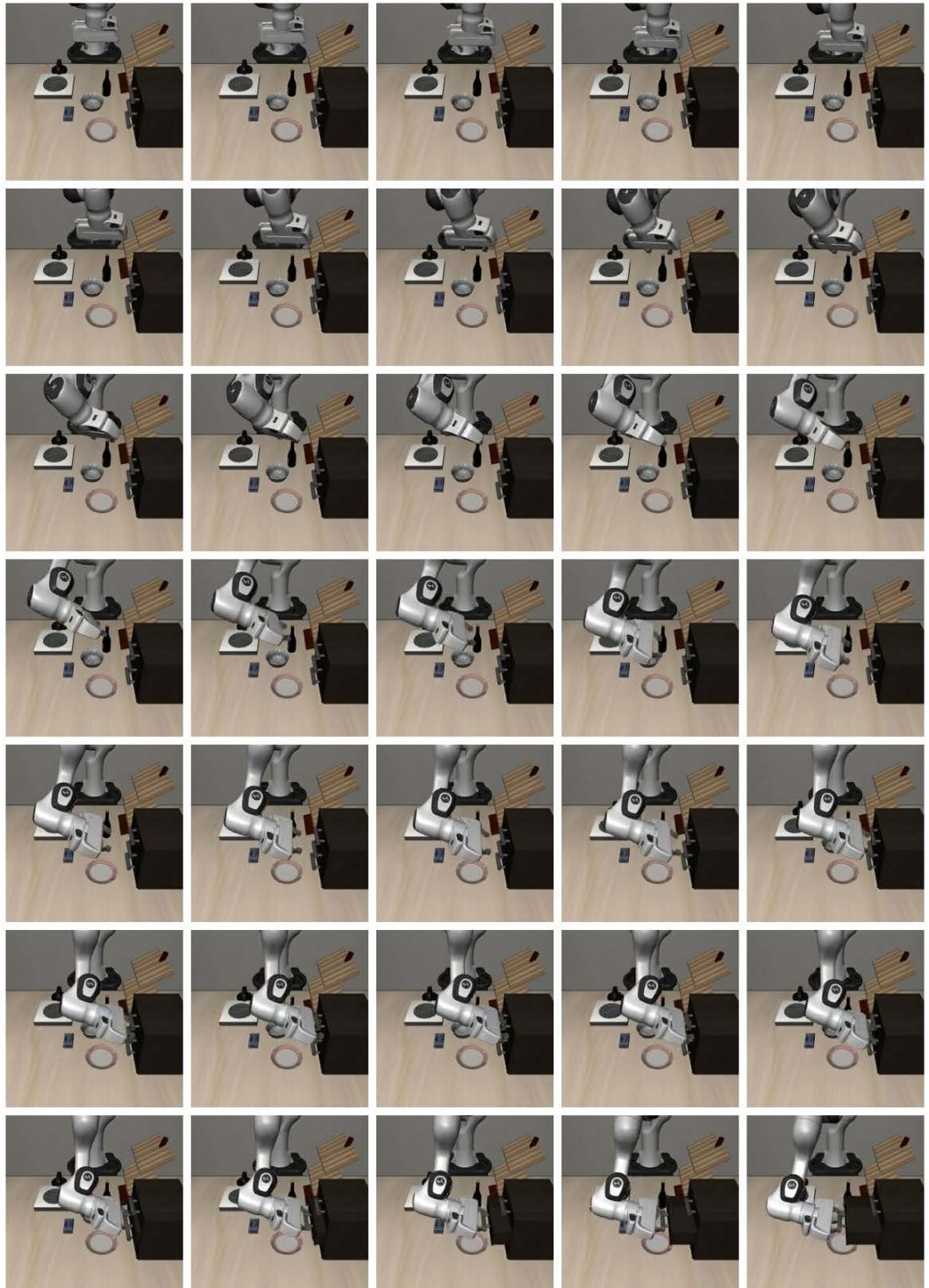

Figure 11: **MetaVLA Execution Sequence Example on LIBERO-Goal.** Instruction: *Open the middle drawer of the cabinet*

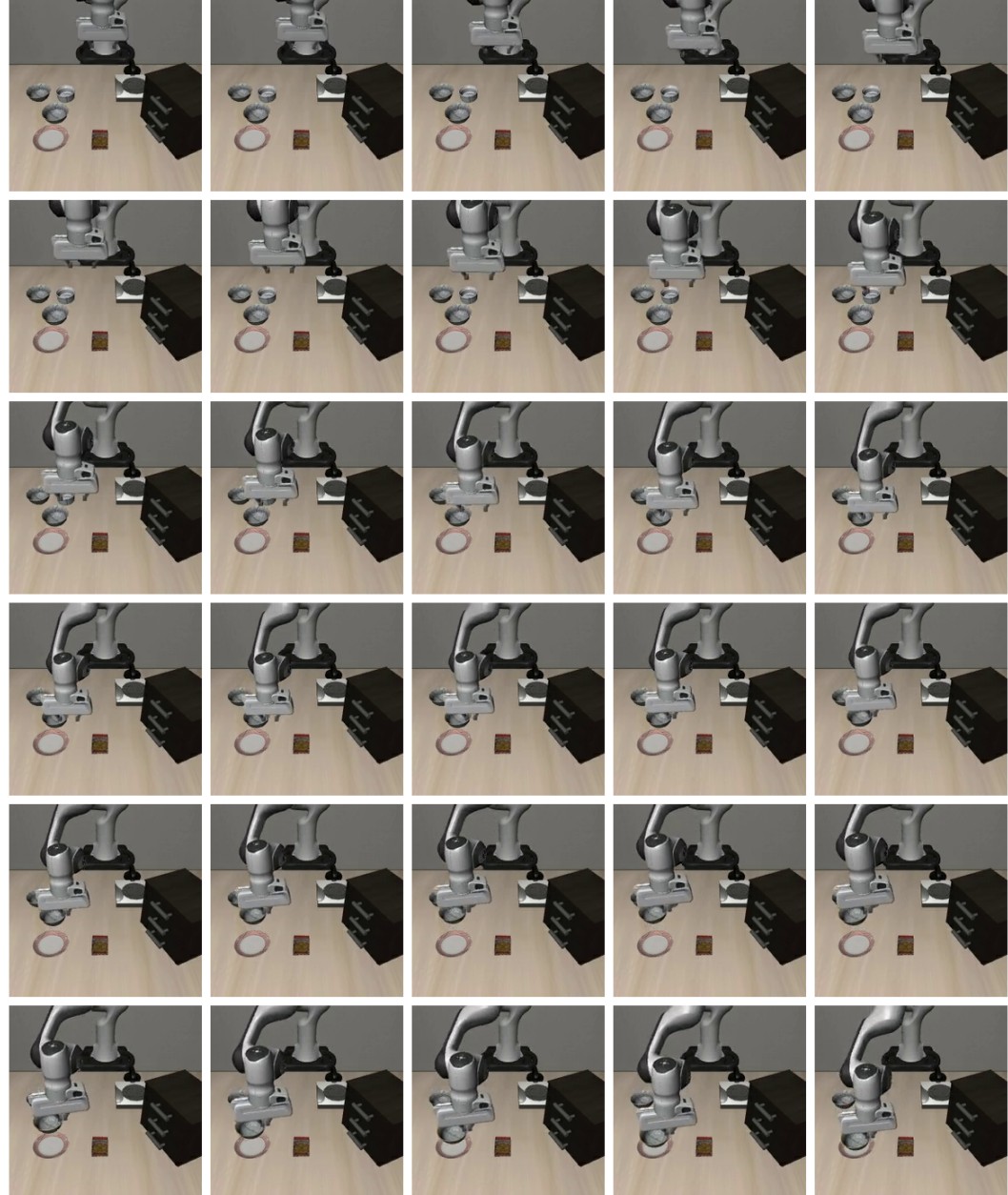

Figure 12: **MetaVLA Execution Sequence Example on LIBERO-Spatial.** Instruction: *Pick up the black bowl between the plate and the ramekin and place it on the plate*

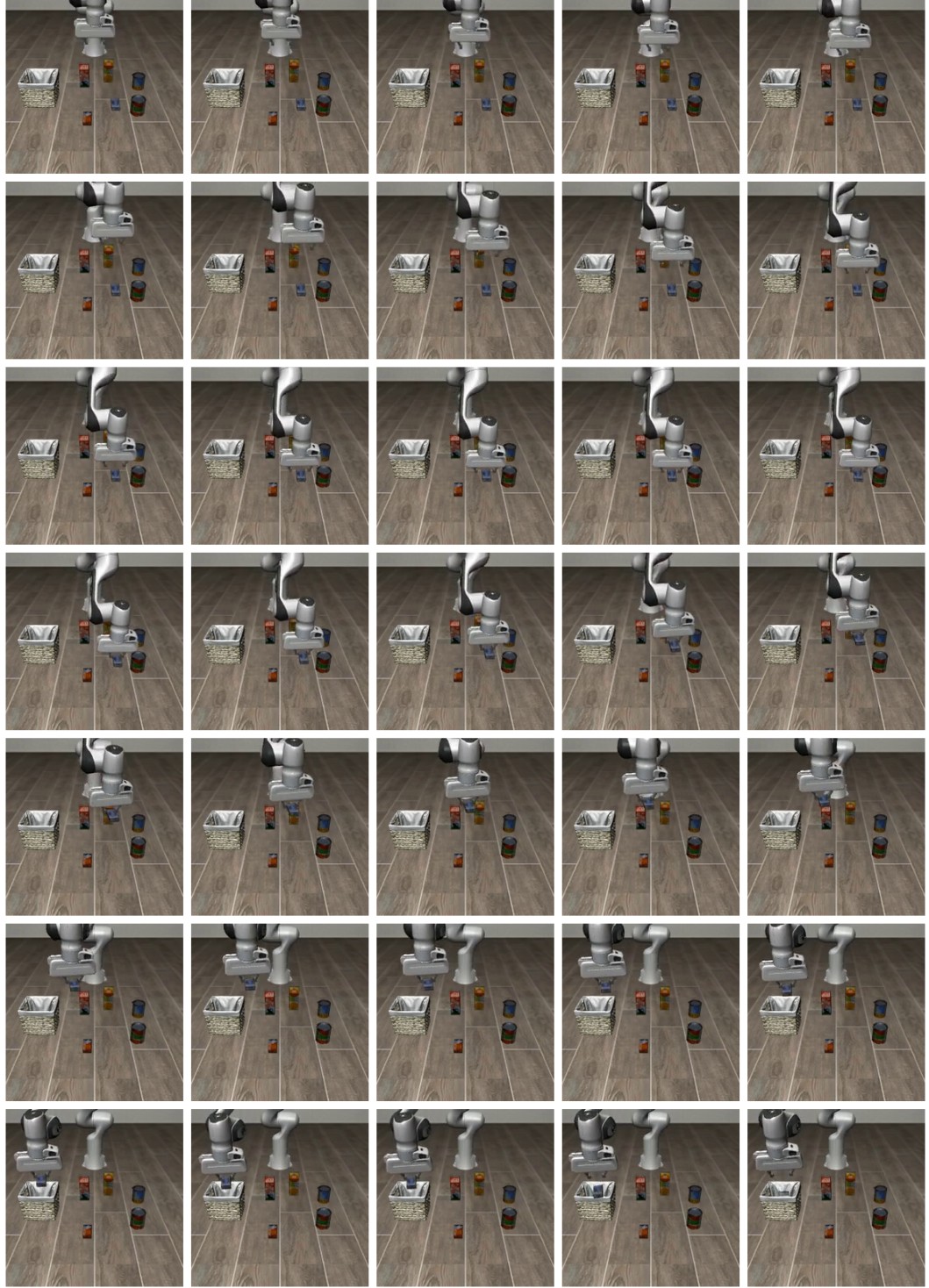

Figure 13: **MetaVLA Execution Sequence Example on LIBERO-Object.** Instruction: *Pick up the cream cheese and place it in the basket*

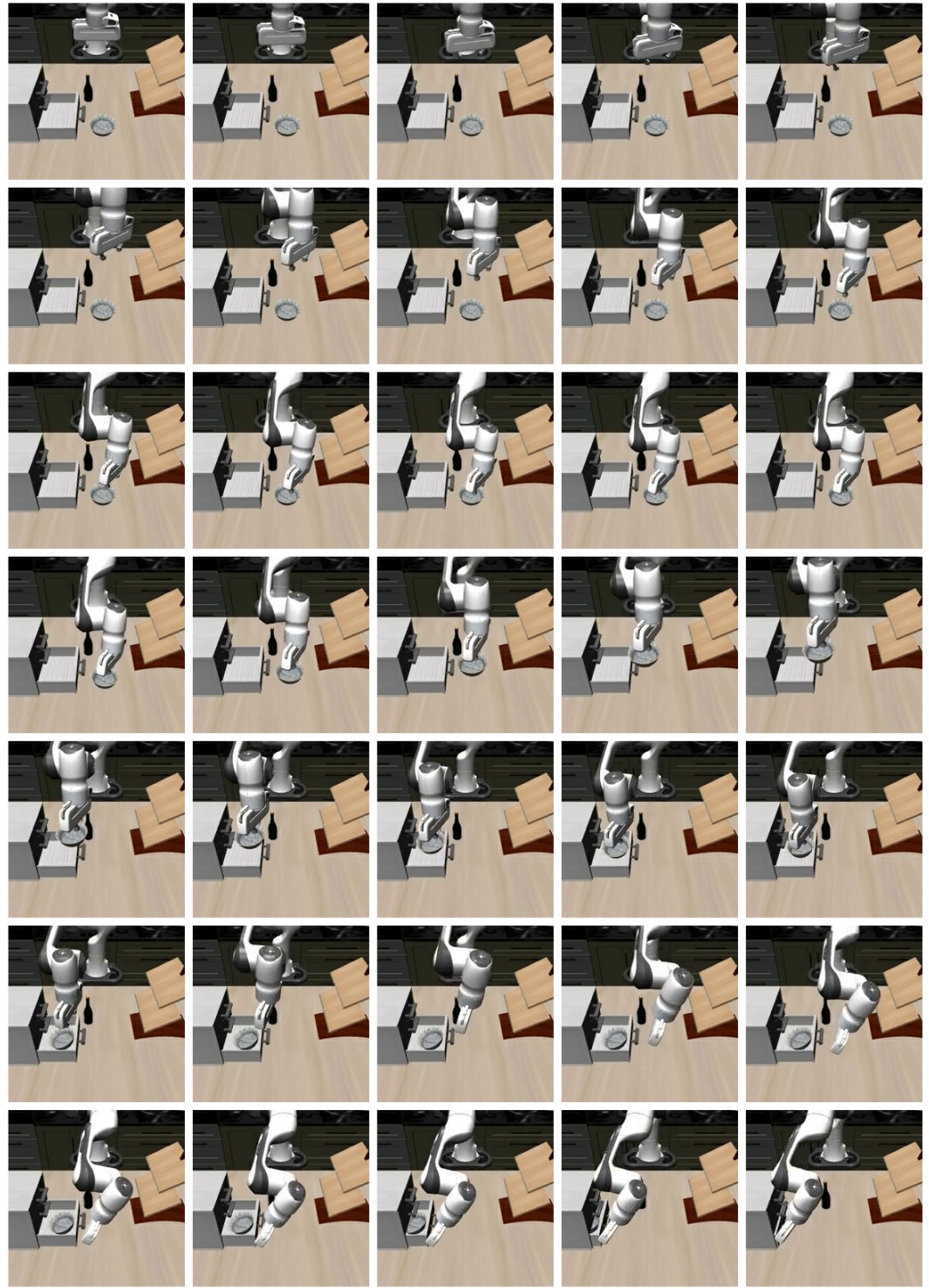

Figure 14: **MetaVLA Execution Sequence Example on LIBERO-Long.** Instruction: *Put the black bowl in the bottom drawer of the cabinet and close it*

| Symbol | Name | Definition |
|--------|------|-----------|
| $\mathbf{x}_T$ | Target feature | Encoded input feature for target queries. |
| $\mathbf{y}_T$ | Target action | Action corresponding to target query $\mathbf{x}_T$. |
| $\mathbf{x}_C$ | Context features | Encoded input features for context queries. |
| $\mathbf{y}_C$ | Context actions | Action corresponding to context query $\mathbf{x}_C$. |
| $(x_{Ci}, y_{Ci})$ | Context pair $i$ | A single feature–action pair from the context bank. |
| $\mathbf{r}_{Ci}$ | Deterministic context rep. | Self-attention rep. for context pair i. |
| $\mathbf{r}_T$ | Deterministic target rep. | Cross-attention rep. of $x_T$ attending to $\{x_{Ci}\}$. |
| $\mathbf{s}_{Ci}$ | Stochastic context rep. | Self-attention rep. used to compute latent posterior. |
| $\bar{\mathbf{s}}_C$ | Mean stochastic context rep. | Averaged stochastic representation over all context rep. $\{\mathbf{s}_{Ci}\}$. |
| $\bar{\mathbf{s}}_T$ | Mean stochastic target rep. | Averaged stochastic rep. over all target rep. $\{\mathbf{s}_{Ti}\}$. |
| $z$ | Latent variable | Stochastic latent sampled from approximate posterior. |
| $q(z\|\bar{\mathbf{s}}_C)$ | Context posterior | Approximate posterior over $z$ conditioned on context. |
| $q(z\|\bar{\mathbf{s}}_T)$ | Target posterior | Training posterior used for variational objective. |
| $p(\mathbf{y}_T\|\mathbf{x}_T, \mathbf{r}_T, z)$ | Decoder likelihood | Conditional distribution predicting target actions. |
| $D_{\mathrm{KL}}(\cdot\|\cdot)$ | KL divergence | Regularizer aligning target and context posteriors. |

Table 7: **Symbol table for Action-ANP and MetaVLA.** *rep.* is abbreviation for *representation*.

