# OpenReview forum: "MetaVLA: Unified Meta Co-Training for Efficient Embodied Adaptation"
_ICLR.cc/2026/Conference — ICLR 2026 Poster_

### Official Review · Reviewer_3qCv · 2025-10-28

**Soundness:** 4
**Presentation:** 3
**Contribution:** 4
**Rating:** 8
**Confidence:** 3

**Summary:**

The paper presents MetaVLA, a unified post-training framework for Vision–Language–Action (VLA) models that introduces Context-Aware Meta Co-Training (CAMC). By integrating a lightweight Action-Attentive Neural Process (Action-ANP) module, the method allows a single model to leverage both in-domain and auxiliary task contexts to achieve efficient multi-task adaptation. Experiments on the LIBERO benchmark show consistent improvements over OpenVLA and vanilla multi-task SFT.

**Strengths:**

1. The paper addresses a highly practical and underexplored problem—efficient post-training of VLA models—with a clear motivation and well-articulated methodology.
2. The proposed Action-ANP module is well integrated into the existing architecture and effectively stabilizes multi-task optimization without increasing inference complexity.
3. The experiments are thorough and convincing, showing clear improvements in both success rate and efficiency with detailed ablations that support the claims.

**Weaknesses:**

1. The method is evaluated only on the OpenVLA-7B backbone, leaving its claimed backbone-agnostic property unverified.
2. The inclusion of large-scale π₀․₅ results in Table 1 could confuse readers, as those models operate under significantly different training regimes.

**Questions:**

N/A

---

> ### Author Response · Authors · 2025-11-25
> **Response to Reviewer 3qCv**
>
> We thank the reviewer for the time and effort in reviewing our paper and for recognizing our work to be practical and well-articulated. We will address your concerns in the following.
>
> > **W1**: The method is evaluated only on the OpenVLA-7B backbone, leaving its claimed backbone-agnostic property unverified.
>
> Please see **General Response 1** above.
>
> > **W2**: The inclusion of large-scale π₀.₅ results in Table 1 could confuse readers, as those models operate under significantly different training regimes.
>
> Thank you for raising the concern. Indeed, we included $\pi_ {0.5} $\[1\] to indicate the current state-of-the-art method. However, it follows a completely different training recipe. It leverages very large-scale pre-training data, including in-house datasets. It also uses multiple experts and diffusion objectives, which differ from our goal of efficient post-training and adaptation.
>
> \[1\] Intelligence, P., Black, K., Brown, N., Darpinian, J., Dhabalia, K., Driess, D., ... & Zhilinsky, U. (2025). $\pi_ {0.5} $: a Vision-Language-Action Model with Open-World Generalization. arXiv preprint arXiv:2504.16054.

---

### Official Review · Reviewer_hdkd · 2025-10-29

**Soundness:** 2
**Presentation:** 1
**Contribution:** 2
**Rating:** 4
**Confidence:** 3

**Summary:**

This paper proposes MetaVLA, a post-training framework for fine-tuning vision-language-action (VLA) models. The approach builds on the Attentive Neural Process (ANP) paradigm, maintaining a context bank of demonstrations drawn from both in-domain and auxiliary tasks. During training, MetaVLA leverages the ELBO objective of ANP to reconstruct action sequences while regularizing them toward the underlying demonstration distributions. The method is evaluated on the LIBERO benchmark across four task suites, where it outperforms the OpenVLA baseline with fewer training steps.

**Strengths:**

1. The paper introduces a formulation for post-training VLA models through a single unified model, reducing the need for per-task fine-tuning.
2. The paper reports both training and inference costs, providing a clearer view of the method’s computational efficiency.

**Weaknesses:**

1. The motivation for few-shot meta-learning is to handle data-scarce scenarios, yet this paper builds its context bank from the entire dataset. Although it is sampled at the size of 32, it means full data access is still required. If the full dataset is available, direct supervised fine-tuning is equally feasible. In addition, the main results show that standard SFT performs only about 3% worse than the proposed method, even without the additional GR00T dataset used by the authors. An experiment of using truly few-shot data could strengthen the paper's arguments.
2. The comparison excludes stronger, more recent fine-tuning baselines such as OpenVLA-OFT [1], which reaches ~97% average success on LIBERO with significantly faster inference, whereas MetaVLA achieves only 79%. A comparison between these 2 methods could clarify the paper's contribution.
3. The claimed generalization is tested only within the four LIBERO suites, which are all visually and semantically similar. Evaluation on cross-benchmark transfer tasks, or more diverse embodied tasks (e.g., RLBench, ManiSkill, FrankaKitchen) would be helpful.
4. The method is described as “backbone-agnostic,” yet experiments are conducted solely on the OpenVLA backbone. No evidence is provided that the approach transfers to other architectures.
5. The presentations need substantial improvements. For example, several citation formats are incorrect; when the author or publication name is not mentioned in the text, citations should appear in parentheses. Also, the definitions of variables such as x, y, s_C are unclear. Figure 2 implies inputs are image observations and text instructions, yet these symbols seem to be something else. The experimental notation (e.g., “SFT-4LIBERO + 5single + 1bimanual”) is not defined in the description/ text, making it difficult to interpret results.

[1] Fine-Tuning Vision-Language-Action Models: Optimizing Speed and Success, RSS 2025.

**Questions:**

Please see my comments above.

---

> ### Author Response · Authors · 2025-11-25
> **Response to Reviewer hdkd (Part 1)**
>
> Thank you for recognizing our efficient and unified post-training framework MetaVLA. We believe there might be some misunderstanding to our main contribution. We will address your concerns in the following.
>
> > **W1**: The motivation for few-shot meta-learning is to handle data-scarce scenarios, yet this paper builds its context bank from the entire dataset. Although it is sampled at the size of 32, it means full data access is still required. If the full dataset is available, direct supervised fine-tuning is equally feasible. In addition, the main results show that standard SFT performs only about 3% worse than the proposed method, even without the additional GR00T dataset used by the authors. An experiment of using truly few-shot data could strengthen the paper's arguments.
>
> **We are not solving a few-shot problem. Our main contributions are:**
> 1. Our paper addresses the VLA post-training problem where we would like to **continuously improve adaptation performance** by introducing extra tasks with **domain diversity**. We propose a framework through the integration of cross-task co-training and intelligently leveraging auxiliary data. With our architecture, we ensure stable optimization by avoiding direct backpropagation on auxiliary data—**a design choice positively noted by reviewers dnnY and 3qCv**—which standard SFT cannot achieve (Table 1: SFT-4LIBERO+5single+1bimanual yields 14.5% vs. General Response 1: NORA-Long-SFT-4LIBERO+5single+1bimanual yields 66.38%).
> 2. Our method is **efficient** (also **positively** noted by **reviewers rKvL and 3qCv**), and it saves training GPU hours by **76%** without introducing extra inference cost.
>
> Moreover, MetaVLA is based on co-training, which means it has access to the full suites of the target LIBERO subset anyway. **Therefore, a few-shot setting enforced on the context bank (only) is by default not rigorous and accurate.** A truly valid few-shot setting would require few-shot samples on the target subset as well \[1\] \[2\].
>
> To further address your concern, as elaborated in Section 3.2.2, we split LIBERO into non-overlapping target and context subsets, and intelligently swap the context batch every 200 iterations for efficiency (Section 3.2.3), meaning the **context does not have access to the full data**. For your information, we find that even using a **fixed  (no swap) LIBERO context bank (a batch of 32)** still yields **comparable performance**.
>
> > **W2**: The comparison excludes stronger, more recent fine-tuning baselines such as OpenVLA-OFT \[3\], which reaches ~97% average success on LIBERO with significantly faster inference, whereas MetaVLA achieves only 79%. A comparison between these 2 methods could clarify the paper's contribution.
>
> OpenVLA-OFT proposed multiple encoding methods and decoding strategies for better action representation. While it primarily studies fine-tuning scenarios, it could be applied to pretraining as well.\
> However, **our method studies a very different problem**. Motivated by the observation that **naive autoregressive SFT** training **fails** on co-training across **diverse datasets**. As a result, we propose a post-training framework that leverages meta-learning to **benefit from relevant yet exotic domains** without direct back-propagation on auxiliary data.\
> We recognize OpenVLA-OFT's excellent results on the various tasks, while we propose a solution for practical VLA usage, how to adapt a model to post-train with several datasets with domain diversity and leverage cross-task dependency to improve the target performance.
>
> ---
> \[1\] Li Fei-Fei, R. Fergus and P. Perona, "One-shot learning of object categories," in IEEE Transactions on Pattern Analysis and Machine Intelligence, vol. 28, no. 4, pp. 594-611, April 2006, doi: 10.1109/TPAMI.2006.79. keywords: {Bayesian methods;Probability density function;Management training;Testing;Image databases;Automotive materials;Rough surfaces;Surface roughness;Layout;Taxonomy;Recognition;object categories;learning;few images;unsupervised;variational inference;priors.}.
> \[2\] Vinyals, O., Blundell, C., Lillicrap, T., Kavukcuoglu, K., & Wierstra, D. (2016). Matching Networks for One‑Shot Learning. arXiv preprint arXiv:1606.04080.
> \[3\] Kim, M. J., Finn, C., & Liang, P. (2025). Fine-tuning vision-language-action models: Optimizing speed and success. arXiv preprint arXiv:2502.19645.

---

> ### Author Response · Authors · 2025-11-29
> **Response to Reviewer hdkd (Part 2)**
>
> > **W3**: The claimed generalization is tested only within the four LIBERO suites, which are all visually and semantically similar. Evaluation on cross-benchmark transfer tasks, or more diverse embodied tasks (e.g., RLBench, ManiSkill, FrankaKitchen) would be helpful.
>
> Please see **General Response 3** above.
>
> > **W4**: The method is described as “backbone-agnostic,” yet experiments are conducted solely on the OpenVLA backbone. No evidence is provided that the approach transfers to other architectures.
>
> Please see **General Response 1** above.
>
> > **W5**: The presentations need substantial improvements. For example, several citation formats are incorrect; when the author or publication name is not mentioned in the text, citations should appear in parentheses. Also, the definitions of variables such as x, y, s_C are unclear. Figure 2 implies inputs are image observations and text instructions, yet these symbols seem to be something else. The experimental notation (e.g., "SFT-4LIBERO + 5single + 1bimanual") is not defined in the description/ text, making it difficult to interpret results.
>
> We thank you for your constructive advice on the paper presentation. We have included a new revision of the paper, please see **General Response 2** above.

---

### Official Review · Reviewer_rKvL · 2025-10-31

**Soundness:** 3
**Presentation:** 3
**Contribution:** 3
**Rating:** 8
**Confidence:** 3

**Summary:**

This paper adopts the ANP approach from the field of meta-learning, effectively addressing the issue in VLA models that requires separate training for tasks with large distribution differences, thereby improving training efficiency and reducing the training costs of multiple models. Furthermore, the MetaVLA approach even enables mutual enhancement across different tasks, which is promising for the general multi-task embodied agents.

**Strengths:**

1.	The paper innovatively applies methods from the meta-learning domain to address the cross-task generalization problem in Vision-Language-Action (VLA) models. While VLA research has long focused primarily on compositional generalization, this work provides a valuable direction for future research on cross-task generalization.
2.	The use of meta-learning significantly reduces computational overhead, enabling a single model trained across multiple task suites to achieve performance comparable to models individually trained for each suite. This component can thus eliminate a large amount of redundant and labour-intensive training.
3.	The core ANP component introduced in the paper is decoupled from the backbone architecture, making it highly modular and extensible, and easily adaptable to various VLA model frameworks.

**Weaknesses:**

1.	Extensive experiments are conducted only on the LIBERO+GT000 setting. However, the method should be evaluated on more diverse cross-task datasets and auxiliary data sources to demonstrate its generalizability better.
2.	The explanation of the ANP method in the paper is difficult to follow. It would be better if there were a more detailed exposition. Additionally, the concept of "Context-Aware Meta Co-Training" is not introduced until the Method section; it should be presented earlier in the paper to aid reader comprehension.
3.	The ablation studies do not address key concerns. For example, under what conditions do subtask co-training mutually enhance each other, and when might they degrade performance? Furthermore, how does the domain similarity of different auxiliary data sources affect task synergy?

**Questions:**

1.	In Table 1, performance with ‘single 3’ is worse than with 1. How does the method scale? When more auxiliary data from diverse sources are used, does performance improve consistently? If so, where might the performance plateau?
2.	It would be valuable for the authors to conduct experiments in embodied settings beyond robotic arm manipulation.
3.	Why does SFT-4LIBERO outperform OpenVLA in Table 1? It is generally believed that individual suite-specific training is necessary for optimal performance, yet even a naive co-training approach via SFT surpasses individually trained models. What explains this result?

---

> ### Author Response · Authors · 2025-11-25
> **Response to Reviewer rKvL (Part 1)**
>
> We thank the reviewer for the time and effort in reviewing our paper and for recognizing the efficiency, extensibility, and adaptability of our work on applying methods from meta-learning to address the generalization problem in VLA models. We will address your concerns in the following.
>
> > **W1**: Extensive experiments are conducted only on the LIBERO+GT000 setting. However, the method should be evaluated on more diverse cross-task datasets and auxiliary data sources to demonstrate its generalizability better.
>
> Please see **General Response 3** above.
>
> > **W2**: The explanation of the ANP method in the paper is difficult to follow. It would be better if there were a more detailed exposition. Additionally, the concept of "Context-Aware Meta Co-Training" is not introduced until the Method section; it should be presented earlier in the paper to aid reader comprehension.
>
> Thank you for raising the concern and providing great suggestions. We have made a revision to the paper, please see **General Response 2**.
>
> > **W3**: The ablation studies do not address key concerns. For example, under what conditions do subtask co-training mutually enhance each other, and when might they degrade performance? Furthermore, how does the domain similarity of different auxiliary data sources affect task synergy?\
> > **Q1**: In Table 1, performance with ‘single 3’ is worse than with 1. How does the method scale? When more auxiliary data from diverse sources are used, does performance improve consistently? If so, where might the performance plateau?
>
> **We address W3 and Q1 together below, as they are related.**
>
> Your observation is insightful. We believe that an **effective context bank** does not rely on any single factor in the data. Instead, **three key dimensions** must be **considered together** to define a context that provides meaningful benefit:
>
> 1. Context data is **unseen** during pretraining: Unseen data provides additional signals to aid its action prediction. As shown in Table 1, by using an already seen pretrain dataset as the context bank (MetaVLA-Pretrained-Context-ONLY 74.4% average success rate), the performance is lower than naive co-trained SFT (SFT-4LIBERO 76.2%).
> 2. Context data shares **relevance** to the target task (domain and embodiment): By sharing some domain similarities, the context could provide good priors to help the model better understand the target task. We add a new experiment, MetaVLA+3bimanual (72.9% average success rate), where two tasks are distant and one is similar to the target. Compared to MetaVLA+1single+1bimanual (78.5%), performance degrades.
> 3. Context data provides **diversity** (domain and embodiment): Aggregating cross-task signals improves generalization. As shown in Table 1, MetaVLA+1single+1bimanual (78.5%) outperforms MetaVLA+3single (78.2%), despite using fewer but more diverse tasks.
>
> We summarize the mentioned experiments as follows:
>
> | Experiment/Condition | Goal (%) | Spatial (%) | Object (%) | Long (%) | Average (%) |
> | :--- | :--- | :--- | :--- | :--- | :--- |
> | SFT-4LIBERO | 77.8 | 84.8 | 87.4 | 54.7 | 76.2 |
> | MetaVLA+1single+1bimanual | 78.5 | 89.0 | 87.4 | 59.0 | 78.5 |
> | MetaVLA-Pretrained-Context-ONLY | 74.4 | 85.4 | 85.4 | 52.3 | 74.4 |
> | MetaVLA+3single | 78.0 | 88.0 | 87.2 | 59.7 | 78.2 |
> | MetaVLA+3bimanual (**newly added**) | 74.0 | 81.5 | 86.5 | 49.5 | 72.9 |

---

> ### Author Response · Authors · 2025-11-25
> **Response to Reviewer rKvL (Part 2)**
>
> > **Q2**: It would be valuable for the authors to.conduct experiments in embodied settings beyond robotic arm manipulation.
>
> The backbone of MetaVLA is pretrained on large‑scale manipulation datasets. Transferring to a new embodiment is an interesting and important topic. We leave it for future research.
>
> > **Q3**: Why does SFT-4LIBERO outperform OpenVLA in Table 1? It is generally believed that individual suite-specific training is necessary for optimal performance, yet even a naive co-training approach via SFT surpasses individually trained models. What explains this result?
>
> 1. Recent studies show that **multi-task co-training** benefits both VLAs\[1\] and MLLMs\[2, 3\], empirically **improving generalization** by exposing models to greater variation and diversity at scale.
> 2. Moreover, the training data from the four LIBERO benchmarks share a **similar domain**. Therefore, the co-training on these data easily benefits from the similar knowledge, state, and action spaces. Our experiments indeed find that vanilla SFT on these similar-domain datasets yields **decent gain**.
> 3. Finally, while naive SFT benefits from in-domain co-training, its performance drops sharply when incorporating out-of-domain data such as GR00T. In contrast, MetaVLA leverages such **heterogeneous data** as a context bank—excluded from backpropagation—which helps boost performance by **avoiding optimization instability**.
>
> Detailed analysis is provided in Section 4.6 **"Why does our method work?"**.
>
> \[1\] Intelligence, P., Black, K., Brown, N., Darpinian, J., Dhabalia, K., Driess, D., ... & Zhilinsky, U. (2025). $\pi_ {0.5} $: a Vision-Language-Action Model with Open-World Generalization. arXiv preprint arXiv:2504.16054. \
> \[2\] Bai, S., Chen, K., Liu, X., Wang, J., Ge, W., Song, S., ... & Lin, J. (2025). Qwen2. 5-vl technical report. arXiv preprint arXiv:2502.13923. \
> \[3\] Grattafiori, A., Dubey, A., Jauhri, A., Pandey, A., Kadian, A., Al-Dahle, A., ... & Vasic, P. (2024). The llama 3 herd of models. arXiv preprint arXiv:2407.21783.

---

### Official Review · Reviewer_dnnY · 2025-11-03

**Soundness:** 2
**Presentation:** 3
**Contribution:** 2
**Rating:** 4
**Confidence:** 3

**Summary:**

This paper introduces MetaVLA, a unified framework designed to improve post-training efficiency and generalization in VLAmodels. Unlike prior work that focuses on larger datasets or architectural changes, MetaVLA targets the post-training stage, addressing the limitations of naive multi-task SFT, which struggles with convergence and degraded performance when tasks differ widely in domain or action space.

MetaVLA introduces Context-Aware Meta Co-Training, which intelligently integrates auxiliary tasks through a memory-augmented mechanism inspired by Attentive Neural Processes. This approach allows the model to leverage out-of-domain information without destabilizing optimization, effectively combining the benefits of multi-task learning and meta-learning. The framework is backbone-agnostic, easy to integrate, and applicable beyond SFT to reinforcement learning setups.

Experiments on the LIBERO benchmark demonstrate MetaVLA’s strong performance and efficiency, compared to OpenVLA and multi-task SFT baselines.

**Strengths:**

1. The paper proposes some empirical originality by integrating Attentive Neural Processes (ANP) into a large-scale VLA architecture, introducing Action-ANP, a compact module that enhances meta-learning for low-data task adaptation. This integration allows the model to capture both global and task-specific semantics through self- and cross-attention, improving convergence and generalization.
2. Context-Aware Meta Co-Training effectively combines in-domain and diverse auxiliary tasks, leveraging an external context bank that broadens the learning signal without destabilizing optimization. The framework is able to train a single scalable model instead of separate ones for each task, and is robust to task diversity, handling heterogeneous data.
3. Experimental results show that MetaVLA scales more robustly, leveraging auxiliary data without encountering optimization instability.
4. Authors perform comprehensive ablations to evaluate effectiveness of design choices in each part.

**Weaknesses:**

1. Since the auxiliary task selection part incorporates additional data, such as GR00T dataset and tasks, I wonder whether performance gain of this co-training paradigm comes in architecture design, i.e. attentive neural process, or better usage of more training data?
2. How generalizable is the proposed ANP architecture?Will this be only applicable to Llama 2 model or any other open source LLMs?
3. I'm a little confused about the efficiency argument in the paper. Since during post-training, more training samples are included and seen by VLA model through auxiliary tasks, it seems that this will increase training cost. Are there statistics about how much time does training/inference time take?

**Questions:**

See weakness

---

> ### Author Response · Authors · 2025-11-25
> **Response to Reviewer dnnY**
>
> We thank the reviewer for the time and effort in reviewing our paper and for recognizing our work in easy integration and effectively boosting performance by intelligently leveraging auxiliary tasks without destabilizing optimization. We will address your concerns in the following.
>
> > **W1**: Since the auxiliary task selection part incorporates additional data, such as GR00T dataset and tasks, I wonder whether the performance gain of this co-training paradigm comes in architecture design, i.e., attentive neural process, or better usage of more training data?
>
> The performance **gain** comes **from multiple factors** in our proposed MetaVLA: the cross-task Co-training, the architecture, as well as effectively using auxiliary data. We have done the **following ablation studies**, which have been **positively noted by reviewers dnnY and 3qCv**, to show that they work in conjunction to bring the performance gain:
> 1. **Gain from Co-training:** In Table 1, we co-trained four tasks as SFT-4LIBERO without architecture change. We have found that under co-training, it has already enhanced the performance (SFT-4LIBERO 76.2% average success rate vs OpenVLA-Individual 74.9%). This indicates that co-training can improve the performance. However, by blindly introducing auxiliary data with domain diversity, the vanilla SFT framework cannot lead to stable training (SFT-4LIBERO+5single+1bimanual 14.5% average success rate)
> 2. **Gain from Architecture**: In Table 1, we show the boost from architecture (MetaVLA 77.8% vs SFT-4LIBERO 76.2% average success rate). Yet the architecture requires proper context; by using the wrong context, the performance degrades (MetaVLA-Pretrained-Context-ONLY 74.4% vs SFT-4LIBERO 76.2% average success rate), because pretrained data does not introduce additional information. We further isolate the effect of architecture from co-training in Table 2.
> 3. **Gain from Auxiliary Data**: Further improvement appears on LIBERO suites (MetaVLA+5single+1bimanual 79.3% vs MetaVLA 77.8%) by introducing appropriate auxiliary data.
>
> > **W2**: How generalizable is the proposed ANP architecture?Will this be only applicable to Llama 2 model or any other open source LLMs?
>
> Please see **General Response 1** above.
>
> > **W3**: I'm a little confused about the efficiency argument in the paper. Since during post-training, more training samples are included and seen by the VLA model through auxiliary tasks, it seems that this will increase training cost. Are there statistics about how much time does training/inference time take?
>
> Our training efficiency—**a merit positively noted by reviewers rKvL and 3qCv**—comes from two aspects: **co-training** and **architecture.** With co-training, we train **one model (75K steps)** for four tasks, efficiently leveraging cross-task signals, while OpenVLA trains 4 models (totally 240K steps); with our architecture, auxiliary tasks are used as the context, and do not participate in gradient descent. In Section 4.5, MetaVLA **reduces training hours by 76%**. For **inference**, in Section 4.6 and Appendix Figure 9, MetaVLA has **negligible overhead** compared to OpenVLA baseline (MetaVLA **5.8ms/tok** vs OpenVLA **5.5ms/tok**).

---

### Author Response · Authors · 2025-11-25
**General Response**

We thank all reviewers for their effort in reviewing and providing constructive and insightful feedback.

**We address common concerns raised by reviewers in this section and provide our responses to each reviewer’s specific feedback following their respective reviews.**

> **General Response 1 (For reviewer dnnY, hdkd, 3qCv)**: Does the proposed MetaVLA generalize to other architectures/backbones?

MetaVLA can be easily generalized to other backbones. We conducted extra experiments using a different backbone NORA-Long \[1\] (NORA-Long - Qwen2.5VL-3B\[6\] vs OpenVLA - Llama2-7B\[7\]). Results are shown below:

| | Goal | Spatial | Object | Long | Average |
| :--- | :---: | :---: | :---: | :---: | :---: |
| NORA-Long \[1\]\[2\]\[3\]\[4\]\[5\] | 85.4 | 90.5 | 95.0 | 70.6 | 85.4 |
| NORA-Long-SFT-4LIBERO | 87.0 | 92.5 | 94.0 | 75.5 | 87.3 |
| NORA-Long-SFT-4LIBERO+5single+1bimanual | 73.6 | 79.5 | 75.2 | 37.2 | 66.4 |
| MetaVLA-NORA-Long (ours) | 90.8 | **96.2** | 96.5 | 77.8 | 90.3 |
| MetaVLA-NORA-Long+5single+1bimanual (ours) | **93.8** | 95.8 | **97.2** | **80.2** | **91.8** |

As in the table, without auxiliary data, MetaVLA outperforms NORA-Long by 4.9% on average and NORA-Long-SFT-4LIBERO by 3.0%. When auxiliary tasks are added, the average success rate further boosts by 6.4% compared to NORA-Long. For **more challenging**
suites—**Goal** and **Long**—the gains are **8.4%** and **9.6%**, respectively. Moreover, consistent
with results using the OpenVLA backbone, MetaVLA-NORA-Long+5single+1bimanual **significantly outperforms** its native SFT counterpart, NORA-Long-SFT-4LIBERO+5single+1bimanual,
by **25.4%**.

In the revised paper, we included the above results in a **new Section** 4.4.1, along with their training curves for accuracy and loss added in Figure 10 of Section A.4.3. These training curves further bolster the **stronger stability** and **robustness** of MetaVLA on a **different backbone**, NORA-Long, as more diverse tasks are added—an observation **consistent** with results using OpenVLA backbone. **Together, these
results clearly demonstrate MetaVLA’s backbone-agnostic ability.**



> **General Response 2**: Paper presentation could be improved.

Thank you for the helpful advice on improving the paper, we have made a new revision based on the reviews as follows:
1. (**For reviewer rKvL**) To improve clarity on ANP, we expanded Section 3.2.1 with new paragraphs elaborating on its role and added a comprehensive symbol table (Appendix Table 6) defining all variables used in the text and figures.
3. (**For reviewer rKvL**) We had originally introduced Context-Aware Meta Co-Training in the Introduction. To further clarify this concept, we have now expanded its definition in the fourth paragraph of the Introduction and added a preview of the proposed method at the beginning of Section 3.2.
4. (**For reviewer hdkd**) We corrected citation formatting to properly use parentheses where appropriate.
5. (**For reviewer hdkd**) While Section 3.2.1 has originally defined all variables, we now also reference this section in the relevant figure caption and include a full variable definition table (Appendix Table 6) for clarity.
6. (**For reviewer hdkd**) We added the definition of "Method+NSingle+Mbimanual" to the Table 1 caption, clarifying that it includes N single-arm and M bimanual (two-arm) auxiliary tasks. These tasks have been originally listed in Appendix Table 3 (Section A.2) and visually illustrated in Figure 8.

> **General Response 3 (For reviewer rKvL, hdkd)**: Evaluating on other cross-task/cross-benchmark datasets.

Cross-benchmark and cross-task generalization—evaluating a fine-tuned model on domains unseen during fine-tuning—is a very interesting and highly important open research direction. While our method improves generalization by leveraging auxiliary data with domain diversity, it is not currently designed for out-of-the-box cross-benchmark transfer, which we are excited to pursue in the future research.

---
\[1\] Hung, C. Y., Sun, Q., Hong, P., Zadeh, A., Li, C., Tan, U., ... & Poria, S. (2025). Nora: A small open-sourced generalist vision language action model for embodied tasks. arXiv preprint arXiv:2504.19854. \
\[2\] https://huggingface.co/declare-lab/nora-long-finetuned-libero-spatial \
\[3\] https://huggingface.co/declare-lab/nora-long-finetuned-libero-goal \
\[4\] https://huggingface.co/declare-lab/nora-long-finetuned-libero-10 \
\[5\] https://huggingface.co/declare-lab/nora-long-finetuned-libero-object \
\[6\] Bai, S., Chen, K., Liu, X., Wang, J., Ge, W., Song, S., ... & Lin, J. (2025). Qwen2. 5-vl technical report. arXiv preprint arXiv:2502.13923. \
\[7\] Touvron, H., Martin, L., Stone, K., Albert, P., Almahairi, A., Babaei, Y., ... & Scialom, T. (2023). Llama 2: Open foundation and fine-tuned chat models. arXiv preprint arXiv:2307.09288.

---

### Author Response · Authors · 2025-12-01
**Paper and Rebuttal Summary (Part 4)**

**Dear Area Chairs,**

Thank you for your time and effort in reviewing our paper and your contribution to the community. We would like to provide a summary about our paper and the rebuttal process to help present a clearer overview of our paper.

**Continuing from the Part 3:**

**Individual Response:**


**Response to Reviewer hdkd**
1) W1: *Concerns about few-shot learning.*

   We addressed this by clarifying **three key misunderstandings**, each with supporting evidence:

   - **Misunderstanding 1:** Our paper does **not** aim to solve a few-shot learning problem. Instead, our **key contributions** are:
     - **Contribution 1:** We propose a novel training framework and architectural design that enables **continuous performance improvement** by incorporating more heterogeneous datasets with **diverse domains** without destabilizing optimization—something standard **SFT fails** to achieve, as shown in our experiments.
     - **Contribution 2:** MetaVLA is **efficiency aware**, which saves training GPU hours by **76%** without introducing extra inference cost.

   - **Misunderstanding 2:** The review's few-shot setting enforced on the context bank (only) is by default **not rigorous and accurate**. A truly valid few-shot setting would require few-shot samples on the **target subset** as well \[1\]\[2\].

   - **Misunderstanding 3:** The claim that our context bank requires access to the full dataset is **incorrect**.
     - We partition the full training set into *target* and *context* subsets (Section 3.2.2) and **intelligently swap** context samples at medium frequency to **maintain** both **effectiveness** and training **efficiency** (Section 3.2.3). This means context does **not** access the full data.
     - Additionally, from our experiments, even using a **fixed LIBERO context bank (batch size 32)** without swapping achieves **comparable performance**, further supporting this point.

2) W2: *Suggestion to compare MetaVLA with OpenVLA-OFT \[3\].*

   We **added the following comparison** between MetaVLA and OpenVLA-OFT:
   - We acknowledge OpenVLA-OFT's strong results and are positive about its potential to extend from fine-tuning to pretraining.
   - However, MetaVLA addresses a **different problem**:
     - **OpenVLA-OFT** introduces improved encoders/decoders for better action representation.
     - **MetaVLA** proposes a highly practical training paradigm along with purposeful architectural design which enables **continuous performance improvements** by leveraging relevant yet diverse auxiliary domains **without destabilizing optimization**—which naïve SFT fails to do.

3) W3: *Suggestion to evaluate on cross-benchmark tasks.*

   Addressed in **General Response 3**.

4) W4: *Request for more evidence of transfer to other architectures.*

   Addressed in **General Response ** with experiments on the NORA-Long (Qwen2.5VL) backbone.

5) W5: *Suggestion to improve presentation.*

   Addressed in **General Response 2**.

---
\[1\] Li Fei-Fei, R. Fergus and P. Perona, "One-shot learning of object categories," in IEEE Transactions on Pattern Analysis and Machine Intelligence, vol. 28, no. 4, pp. 594-611, April 2006, doi: 10.1109/TPAMI.2006.79. keywords: {Bayesian methods;Probability density function;Management training;Testing;Image databases;Automotive materials;Rough surfaces;Surface roughness;Layout;Taxonomy;Recognition;object categories;learning;few images;unsupervised;variational inference;priors.}.
\[2\] Vinyals, O., Blundell, C., Lillicrap, T., Kavukcuoglu, K., & Wierstra, D. (2016). Matching Networks for One‑Shot Learning. arXiv preprint arXiv:1606.04080.
\[3\] Kim, M. J., Finn, C., & Liang, P. (2025). Fine-tuning vision-language-action models: Optimizing speed and success. arXiv preprint arXiv:2502.19645.


**Response to Reviewer 3qCv**
1) W1: *More evidence for the claim of backbone-agnostic is needed.*

   Addressed in **General Response 1** with experiments on the NORA-Long (Qwen2.5VL) backbone.

2) W2: *The inclusion of large-scale π₀.₅ results in Table 1 could confuse readers, as those models operate under significantly different training regimes.*

   We clarify that we included $\pi_ {0.5} $\[1\] solely to indicate the current state-of-the-art. We fully agree with the reviewer and acknowledge that it operates under a significantly different training recipe, and thus it is not meant for direct comparison.

---
\[1\] Intelligence, P., Black, K., Brown, N., Darpinian, J., Dhabalia, K., Driess, D., ... & Zhilinsky, U. (2025). $\pi_ {0.5} $: a Vision-Language-Action Model with Open-World Generalization. arXiv preprint arXiv:2504.16054.

---

### Author Response · Authors · 2025-12-01
**Paper and Rebuttal Summary (Part 3)**

**Dear Area Chairs,**

Thank you for your time and effort in reviewing our paper and your contribution to the community. We would like to provide a summary about our paper and the rebuttal process to help present a clearer overview of our paper.

**Continuing from the Part 2:**

**Individual Response:**


**Response to Reviewer dnnY**
1) W1: *Which part does the performance gain come from?*
We addressed this by identifying following **three key factors**, whose combined effect contributes to the observed performance gain:
   - Cross-task Co-Training
   - Architecture design
   - Effective use of auxiliary data
For each factor, we provided **supporting ablation results** from the paper

2) W2: *Can MetaVLA generalize to other open-source LLMs?*
Addressed in **General Response 1** with experiments on the NORA-Long (Qwen2.5VL) backbone.

3) W3: *Why is training more efficient? Are there statistics on training/inference time?*
We clarified this by:
   - Analyzing two main efficiency sources: **co-training** and **architecture**
   - Providing training efficiency statistics (originally reported in the paper):
     - **Figure 1(b)** and **Table 1**: MetaVLA trains **1 model (75K steps)** for 4 tasks vs. OpenVLA’s 4 models and **240K steps**
     - **Section 4.5**: MetaVLA reduces GPU training hours by **76%**
   - Reporting inference efficiency (**Section 4.6**, Appendix **Figure 9**):  MetaVLA shows **negligible overhead** compared to OpenVLA (**5.8 ms/token** vs. **5.5 ms/token**)


**Response to Reviewer rKvL**
1) W1: *Suggestion to evaluate on cross-task datasets.*

   Addressed in **General Response 3**.

2) W2: *Suggestion to improve presentation.*

   Addressed in **General Response 2**.

3) W3 and Q1: *What is the relationship between subtask co-training / different contexts and performance change?*

   We addressed W3 and Q1 together, as they are closely related.
   We analyzed three key properties of an **effective context bank**:
      - **Unseen** in the pretraining dataset
      - **Relevant** to the target task domain
      - Rich in **domain diversity** to provide complementary learning signals for generalization

   To support this, we presented **paired experimental comparisons** for each dimension, where one experiment (**MetaVLA+3bimanual**) was **newly added**, and the rest were already included in the original paper.

4) Q2: *Suggestion to go beyond robotic arm manipulation.*

   MetaVLA’s backbone is pretrained on large-scale manipulation datasets. Transferring to a new embodiment is an exciting and important direction, which we leave for future work.

6) Q3: *Why is multi-task co-training better than single-task finetuning?*

   We provided the rationale that **multi-task co-training** improves generalization in both **VLAs** \[1\] and **MLLMs** \[2, 3\].
   We further referred back to our experiments which **support this benefit across**:
      - Tasks with similar domains
      - Heterogeneous tasks with greater domain diversity

---
\[1\] Intelligence, P., Black, K., Brown, N., Darpinian, J., Dhabalia, K., Driess, D., ... & Zhilinsky, U. (2025). *π₀.₅: A Vision-Language-Action Model with Open-World Generalization*. arXiv:2504.16054
\[2\] Bai, S., Chen, K., Liu, X., Wang, J., Ge, W., Song, S., ... & Lin, J. (2025). *Qwen2.5-VL Technical Report*. arXiv:2502.13923
\[3\] Grattafiori, A., Dubey, A., Jauhri, A., Pandey, A., Kadian, A., Al-Dahle, A., ... & Vasic, P. (2024). *The LLaMA 3 Herd of Models*. arXiv:2407.21783

---

### Author Response · Authors · 2025-12-01
**Paper and Rebuttal Summary (Part 2)**

**Dear Area Chairs,**

Thank you for your time and effort in reviewing our paper and your contribution to the community. We would like to provide a summary about our paper and the rebuttal process to help present a clearer overview of our paper.

**Continuing from the Part 1:**

We *summarize our rebuttal responses*. It consists of **General Response** and **Individual Response**:

**General Response**

**General Response 1 (For reviewers dnnY, hdkd, 3qCv)**: *Does MetaVLA generalize to other backbones/architectures?*

We addressed this by adding a new experiment, experiment insights and analysis, and corresponding revisions in **Section 4.4.1** and **Figure 10** (Appendix A.4.3) of the updated paper.

1) New Experiment Added:
We integrated **MetaVLA** into a different backbone, **NORA-Long** \[1\]:
- Backbone difference:
  - NORA-Long → *Qwen2.5VL-3B* \[2\]
  - OpenVLA → *LLaMA2-7B* \[3\]
- NORA-Long was chosen for it is a stronger LIBERO baseline than the NORA variant.

2) Results and Insights of MetaVLA on NORA-Long:
- MetaVLA **improves average success rates** both with and without auxiliary data.
- Gains on **challenging** suites—**Goal** and **Long**—are **+8.4%** and **+9.6%**, respectively
- MetaVLA-NORA-Long+5single+1bimanual **outperforms** its vanilla SFT counterpart by **+25.4%**.

3) Paper Revisions Provided:
- Results added to **Section 4.4.1**
- Training curves (accuracy/loss) newly included in **Figure 10** in Appendix **Section A.4.3**, further showing MetaVLA’s **stability** and **robustness** across backbones

Together these results **clearly demonstrate MetaVLA’s backbone-agnostic generalization.**

**General Response 2 (For reviewers rKvL and hdkd):** *Paper presentation could be improved.*

We addressed the concerns as follows:
1) We clarified the concept of ANP (for Reviewer rKvL)
- Expanded **Section 3.2.1** with a more detailed explanation
- Added a comprehensive symbol table (**Appendix Table 6**) defining all variables in ANP formulas

2) We clarified Context-Aware Meta Co-Training (for Reviewer rKvL)
- Expanded its definition in **Paragraph 4 of the Introduction**, which previously provided a more brief description
- Added a brief method preview at the start of **Section 3.2**

3) Other presentation fixes (for Reviewer hdkd)
- Corrected citation formatting.
- Improved variable clarity:
  - Referenced **Section 3.2.1** in relevant figure captions
  - Included all variable definitions in a **newly added Appendix Table 6**
- Clarified experimental notation:
  - Updated **Table 1** caption to define “Method+NSingle+Mbimanual” as including *N* single-arm and *M* bimanual (two-arm) auxiliary tasks
  - These tasks are originally listed in **Appendix Table 3 (Section A.2)** and visualized in **Figure 8** for reference.

**General Response 3 (For reviewers rKvL, hdkd):** *Evaluating on cross-task/cross-benchmark datasets is suggested.*

We acknowledge the value of this direction and clarify its distinction from our current scope, while expressing strong interest in future exploration. Details below:

Cross-task and cross-benchmark generalization—evaluating on domains unseen during fine-tuning—is an important and promising area. While our method enhances generalization via auxiliary data with diverse domains, it is not yet designed for out-of-the-box cross-benchmark transfer. We are enthusiastic about pursuing this direction in future work.

---
\[1\] Intelligence, P., Black, K., Brown, N., Darpinian, J., Dhabalia, K., Driess, D., ... & Zhilinsky, U. (2025). $\pi_ {0.5} $: a Vision-Language-Action Model with Open-World Generalization. arXiv preprint arXiv:2504.16054. \
\[2\] Bai, S., Chen, K., Liu, X., Wang, J., Ge, W., Song, S., ... & Lin, J. (2025). Qwen2. 5-vl technical report. arXiv preprint arXiv:2502.13923. \
\[3\] Grattafiori, A., Dubey, A., Jauhri, A., Pandey, A., Kadian, A., Al-Dahle, A., ... & Vasic, P. (2024). The llama 3 herd of models. arXiv preprint arXiv:2407.21783.

---

### Author Response · Authors · 2025-12-01
**Paper and Rebuttal Summary (Part 1)**

**Dear Area Chairs,**

Thank you for your time and effort in reviewing our paper and your contribution to the community. We would like to provide a summary about our paper and the rebuttal process to help present a clearer overview of our paper.

**TL;DR**
MetaVLA introduces a novel, backbone-agnostic training framework with architectural enhancements that enables **continuous adaptation performance gains** by leveraging heterogeneous datasets with domain diversity—**without destabilizing optimization**, a limitation of vanilla SFT. It also delivers strong **efficiency benefits**, reducing the number of models to train to **one** and cutting GPU training hours by **76%**, all **without adding inference overhead**.

We *summarize strengths* across all reviews as below:
1. **Strength 1 – Well-Motivated Problem (by Reviewers rKvL and 3qCv):**
   MetaVLA addresses a highly practical and underexplored problem—efficient post-training of VLA models—which also lays the groundwork for valuable future research in cross-task generalization.
2. **Strength 2 – Methodological Novelty (by Reviewers dnnY and 3qCv):**
   MetaVLA introduces a novel training framework whose full design enables continuous adaptation performance gains without destabilizing optimization.
3. **Strength 3 – Efficiency Merit (by Reviewers rKvL, hdkd, and 3qCv):**
   MetaVLA significantly reduces training costs—fewer models and GPU hours, without increasing inference complexity.
4. **Strength 4 – Robust Experiments and Ablations (by Reviewers dnnY and 3qCv):**
   Extensive experiments demonstrate scalable improvements and efficiency. Comprehensive ablations validate the effectiveness of individual design components and support the claims.
5. **Strength 5 – Backbone-Agnostic, Modular, and Extensible (by Reviewer rKvL):**
 MetaVLA is highly modular and extensible, and easily adaptable to various VLA model frameworks.

---

### Meta-Review · Area_Chair_mi46 · 2026-01-05

**Summary:**

This paper presents a post-training framework for VLA models that addresses a practical bottleneck in embodied adaptation, i.e., the cost and scalability of task-specific fine-tuning. While the initial submission raised reasonable concerns regarding attribution of gains, backbone generality, efficiency accounting, and clarity of presentation, the rebuttal substantially strengthens the work by adding convincing cross-backbone experiments, clarifying the role of ANP-based meta co-training, and providing concrete statistics on training and inference efficiency. While some limitations remain (e.g., the lack of comparison to the strongest contemporaneous fine-tuning baselines and evaluation beyond the LIBERO benchmark), these do not undermine the core contribution or empirical validity of this work. Based on the review ratings and rebuttal responses, the AC believes the paper is clearly above the acceptance threshold, which should be of interest to the embodied AI community.

**Reviewer Concerns:**

1. Attribution of Performance Gains (Data vs. Method?): Without controlled comparisons (same data, different training paradigms), the core methodological contribution is not conclusively validated.
2. Backbone-Agnostic Claim Is Under-Supported: This weakens a central claim of the paper.
3. Lack of Clarity in Efficiency Argument
4. Meta-Learning Justification Is Conceptually Weak (not true few-shot setting)
5. Insufficient Baseline Comparisons & Limited Generalization

The rebuttal substantially addresses most of the decisive technical concerns, particularly those that drove the borderline rejects (R1, R3).

**Reviewer Scores:**

For R1, the question of whether gains come from architecture or additional data is mitigated by explicit ablations and by framing MetaVLA’s improvement as the combined effect of co-training, ANP-based adaptation, and auxiliary data rather than a single factor. Then, the generalizability concern is addressed by adding new experiments on a different backbone (NORA-Long with Qwen2.5VL). Finally, the efficiency confusion is clarified with precise statistics, which show that, despite using auxiliary tasks, MetaVLA reduces total training steps and GPU hours through unified co-training and incurs negligible inference overhead, making the efficiency argument transparent and credible. The AC believes that, if R1 had been able to participate in the discussion, the rating would have been increased to 6 or above.

As for R3, the AC feels his concerns are partially addressed. The rebuttal strengthens the paper by providing evidence for the generality of the backbone and enhancing the method's clarity. However, the core concern of the meta-learning formulation is not fully addressed, as it assumes full data access, and the fact that standard SFT already performs competitively remains only reframed rather than directly refuted (no genuine few-shot or data-scarce experiment is conducted). While the authors provide further clarification, it remains unclear whether R3 would be satisfied since no further discussion is allowed. In addition, the lack of comparison with stronger baselines such as OpenVLA-OFT and the absence of cross-benchmark evaluation persist, meaning R3’s concerns are mitigated but not decisively resolved. With such responses, the AC feels that R3 might remain the original ratings (or be increased to 6 at most).

---

### Decision · Program_Chairs · 2026-01-26

Accept (Poster)